# Successor-like representation guides the prediction of future events in human visual cortex and hippocampus

Matthias Ekman*, Sarah Kusch, Floris P de Lange

Radboud University Nijmegen, Donders Institute for Brain, Cognition and Behaviour, Nijmegen, Netherlands

**Abstract** Human agents build models of their environment, which enable them to anticipate and plan upcoming events. However, little is known about the properties of such predictive models. Recently, it has been proposed that hippocampal representations take the form of a predictive map-like structure, the so-called successor representation (SR). Here, we used human functional magnetic resonance imaging to probe whether activity in the early visual cortex (V1) and hippocampus adhere to the postulated properties of the SR after visual sequence learning. Participants were exposed to an arbitrary spatiotemporal sequence consisting of four items (A-B-C-D). We found that after repeated exposure to the sequence, merely presenting single sequence items (e.g., - B - -) resulted in V1 activation at the successor locations of the full sequence (e.g., C-D), but not at the predecessor locations (e.g., A). This highlights that visual representations are skewed toward future states, in line with the SR. Similar results were also found in the hippocampus. Moreover, the hippocampus developed a coactivation profile that showed sensitivity to the temporal distance in sequence space, with fading representations for sequence events in the more distant past and future. V1, in contrast, showed a coactivation profile that was only sensitive to spatial distance in stimulus space. Taken together, these results provide empirical evidence for the proposition that both visual and hippocampal cortex represent a predictive map of the visual world akin to the SR.

*For correspondence:
matthias.ekman@donders.ru.nl

## Editor's evaluation

In this paper, Ekman and colleagues present compelling fMRI evidence from a visual sequence task that both the early visual cortex (V1) and the hippocampus represent perceptual sequences in the form of a predictive "successor" representation, where the current state is represented in terms of its future (successor) states in a temporally discounted fashion. In both brain structures, there was evidence for upcoming, but not preceding steps in the sequence, and these results were found only in the temporal but not spatial domain. This study offers the fundamental suggestion that both the hippocampus and V1 represent temporally structured information in a predictive, future-oriented manner.

## Introduction

Anticipation and planning of future visual input require knowledge of the relational structure between events. The relational structure, for instance that stimulus B usually follows stimulus A, is learned through exposure during past experiences (*Behrens et al., 2018*; *Finnie et al., 2021*; *Gavornik and Bear, 2014*) and can be used to build a model or cognitive map (*Tolman, 1948*) that enables us to generate inferences in situation with noisy or partial input (*Ekman et al., 2017*; *Momennejad, 2020*; *Schwartenbeck et al., 2021*).

In the visual domain, with rapidly changing input, it remains unknown what the inherent properties of the model underlying our predictions are. On the one hand, such a model needs to be efficient enough to generate predictions from a constant stream of visual input, while on the other hand, it also allows for flexible updating in an ever-changing environment. In the context of hippocampal representations, the successor representation (SR) has been recently proposed (*Dayan, 1993*; *Stachenfeld et al., 2017*) to combine the trade-off between both flexible and efficient model properties. The SR postulates a predictive representation in which the current state is represented in terms of its future (successor) states, in a temporally discounted fashion. The SR is dependent on the actual experience, with states experienced more frequently being represented more strongly. This enables learning in an environment without explicit reward (*Gläscher et al., 2010*). This hypothesis captures many aspects of empirical hippocampal place cell firing pattern, like the exponential decay toward distant future locations (*Alvernhe et al., 2011*; *Mehta et al., 2000*).

Previous research has repeatedly shown that prior expectations influence neural activity in the visual cortex (*Ekman et al., 2017*; *Gavornik and Bear, 2014*; *Hindy et al., 2016*; *Kok et al., 2012*; *Xu et al., 2012*). It remains, however, unknown if SR-like representations are present outside the hippocampus in areas like the early visual cortex (V1) that have a strong retinotopic organization. Theoretically, it is possible that V1 receptive fields, analogous to hippocampal place fields, become tuned to respond not only to the current input, but also to expected future inputs. Here, we propose that the computationally efficient and flexible properties of the SR could in theory also underlie the anticipation of future events in V1.

To directly test this hypothesis, we conducted a functional magnetic resonance imaging (fMRI) study in which participants were presented with an arbitrary visual dot sequence (A-B-C-D). After initial sequence exposure, we introduced occasional omission trials, where only one element of the sequence was presented (e.g., B), while the rest of the sequence (e.g., A, C, and D) was omitted. These partial sequence trials allowed us to study expectations of future stimulus sequences in the absence of physical stimulation. This design allowed us to test the specific assumptions of the SR and also assess whether V1 predictions were better described by an alternative mechanism called pattern completion. Pattern completion describes a framework in which autoassociative connections within

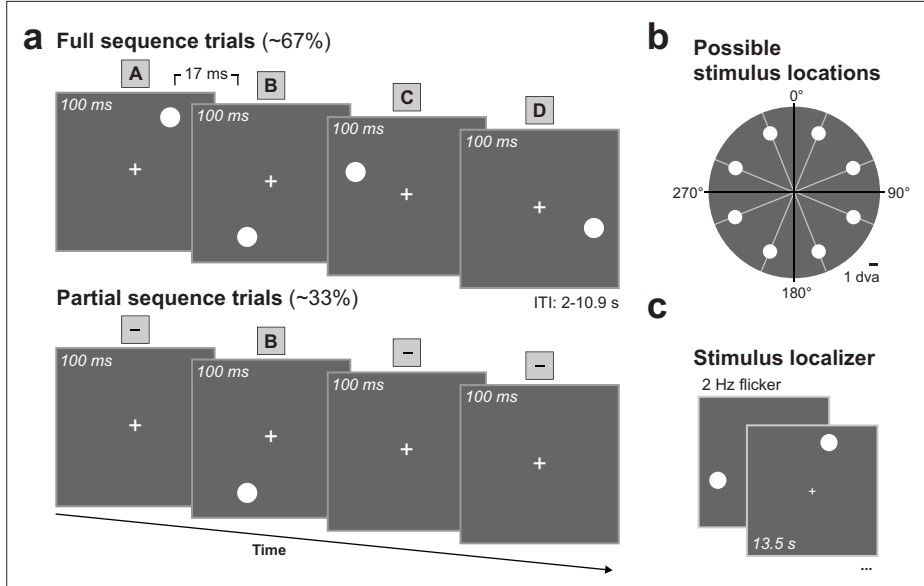

**Figure 1.** Sequence paradigm to probe successor-like representations. (**a**) Stimulus timing for full sequence trials (*top*) and partial sequence trials (*bottom*). During full sequence trials, four dots were presented in rapid succession in a fixed sequence order (A-B-C-D). During partial sequence trials, only one of the four dots was presented, omitting the remaining sequence dots. Here shown for -B - -, while A- - -, - -C-, and - - -D partial trials were also presented. (**b**) Sequences were randomized across subjects such that sequence locations were sampled from a total of eight possible locations with the constraint that every quadrant was stimulated once. Dot locations were evenly spaced around central fixation at a radius of 7 degrees visual angle (dva). (**c**) Independent stimulus localizer trials to map out stimulus representations.

the hippocampal CA3 regions reactivate related sequence items from partial input (*Deuker et al., 2014*; *Leutgeb and Leutgeb, 2007*; *Rolls, 2013*) that is then propagated to sensory regions such as V1 (*Hindy et al., 2016*). In contrast to the SR, pattern completion predicts reactivations of all associated items, without any skewing toward future locations or temporal discounting of events that are farther in the future.

Using fMRI, we found reactivations of future sequence locations (e.g., C-D), but not of past locations (e.g., A) in both V1 and hippocampus. In line with the SR, a model comparison confirmed that predictive representations constitute a map-like structure, with exponential decay toward distant future states. Further, more detailed analysis of predictive codes revealed that hippocampus represented visual locations based on their temporal proximity within the sequence, rather than spatial distance.

Taken together, these data suggest that humans predict upcoming visual input by using a generative model whose properties resemble the SR. Importantly, the presence of SR-like representations in V1 indicates that SR might be a more ubiquitous coding schema that is present beyond hippocampal place cells. Finally, while SR-like representations were found to be present in both V1 and hippocampus, the predictive codes between these areas revealed complementary tuning properties, with hippocampus being sensitive to temporal distance and V1 being more sensitive to retinotopic spatial distance.

## Results

Human observers (N = 35) were exposed to four dots presented in rapid succession that formed an arbitrary visual sequence A-B-C-D (*Figure 1A*). Dot locations were sampled from eight locations (*Figure 1B*) and the resulting possible sequences were randomly assigned across subjects. After an initial exposure period with the full sequence (352 trials outside the scanner, 160 trials inside the scanner), occasionally only one item of the sequence was presented, omitting the remaining sequence items (e.g., partial sequence trial '- B - -' where B is shown and A, C, and D are omitted; *Figure 1A*). Participants were instructed to maintain fixation throughout the experiment, and tasked to detect a slight temporal onset delay (170 ms vs. 17 ms) of the last sequence dot that occurred in ~40% of the

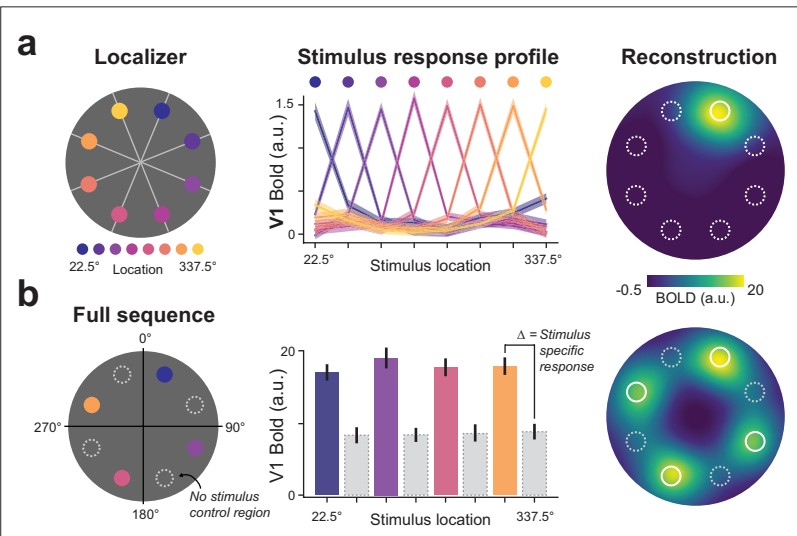

**Figure 2.** V1 stimulus mapping. (**a**) An independent stimulus localizer was used to identify V1 subpopulations that respond to individual dot locations (*left*). Stimulus-response profiles show tuning properties for selected V1 populations (*middle*). Visualizing stimulus activity by projecting group averaged BOLD activity (n=35) into stimulus space (*right*) shows focal activity at the stimulated location with minimal spreading to neighboring locations. (**b**) Identified V1 subpopulations during full sequence trials (left) show heightened BOLD activity compared to non-stimulated control locations (*middle*). Group averaged (n=7) sequence activity projected into stimulus space shows spatially specific activity at the stimulated locations (*right*).

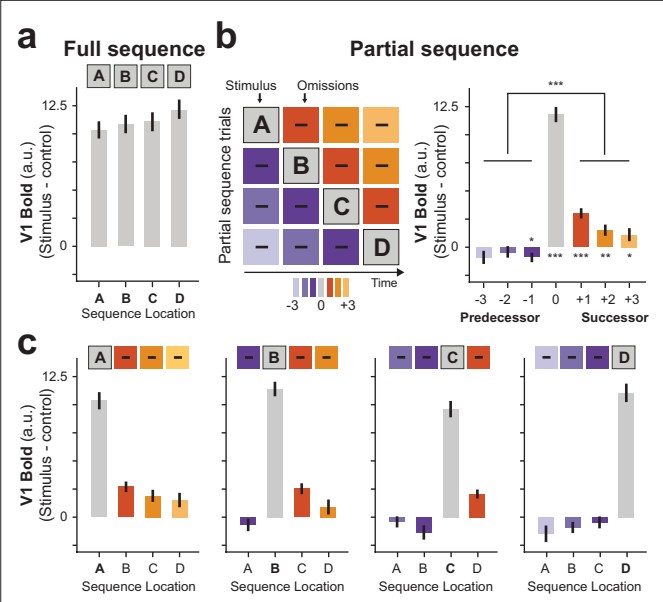

**Figure 3.** Successor-like representation of future sequence events in V1. (**a**) BOLD activity during full sequence trials. (**b**) Schematic of all partial sequence trials (*left*) illustrating the omission of different predecessor (purple), or successor (orange) sequence locations. Group averaged (n=35) V1 activity during partial sequence trials (*right*) shows enhanced activation of successor locations compared to predecessor locations. (**c**) Group averaged V1 activity for individual partial sequence trials. Error bars denote ± s.e.m.; two-tailed t test, \*\*\*p < 0.001; \*\*p < 0.01; \*p < 0.05, uncorrected for multiple comparisons.

full sequence trials. The task was designed to be demanding (group averaged hit rate = 78%, SD = 8%) and to keep participants' attention on the sequence.

We hypothesized that presenting only one item of the sequence would elicit anticipatory activity at the omitted sequence locations that followed the presented stimulus (i.e., successor states), but not at the sequence location that preceded the sequence item (i.e., predecessor states). For instance, during partial '- B - -' trials, we expected activity at omitted sequence locations C (+1) and D (+2), but not at omitted location A (–1).

## Stimulus sequences elicit spatially specific responses in V1

To test our prediction, we first selected V1 sub regions of interest (ROIs) that responded selectively to the eight stimulus locations based on an independent localizer session (*Figure 1C*). Stimulus-response profiles of these eight (retinotopic) ROIs show little coactivation of neighboring locations in the visual field which allows for a precise investigation of location-specific activity (*Figure 2A*). Unsurprisingly, during full sequence trials, BOLD activity at the sequence locations receiving bottom-up visual input was markedly enhanced compared to non-stimulated control locations (*Figure 2B*). Population-based receptive field (pRF) data that was acquired for a subset of participants confirmed that the selected voxels correspond to the retinotopic stimulus locations as expected. For all analyses, we subtracted the average BOLD activity of all control locations from the sequence location activity (*Figure 3A*), which provides an accurate measure of stimulus-specific responses independent of global signal fluctuations for instance due to attention.

## Anticipated stimulus sequences in V1

Briefly flashing individual dots during partial sequence trials, while omitting the other dots of the sequence, allowed us to probe anticipatory activity at the successor and predecessor locations (*Figure 3B*). In line with our predictions, V1 BOLD activity was indeed enhanced at the non-stimulated successor locations compared to the non-stimulated predecessor locations (averaged across all partial trials and sequence locations; t(34) = 6.45, p = 2.23 × 10⁻⁷). The same pattern of future-directed prediction was also evident from the visual inspection of BOLD activity for all partial sequence trials

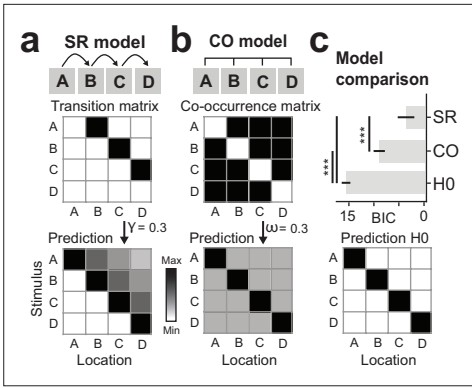

**Figure 4.** Model comparison favors successor-like representation in V1. (**a**) Probing predictions of the successor representation (SR) against the competing co-occurrence (CO) model. The relational structure of the full sequence A-B-C-D is translated into a transition matrix (*top*), where a non-zero value indicates a transition between two states in the sequence. The SR matrix (*bottom*) is computed from the transition matrix, here shown with a temporal discount factor of γ = 0.3 (see Materials and methods). (**b**) The relational structure in the CO model is non-directional, resulting in a constant prediction of past and future states weighted by a factor $\omega$. (**c**) Competing model predictions were fitted to partial sequence trial V1 data of each individual participant with γ and $\omega$ as free parameters. Comparison of model errors showed that the data is most in line with the SR. A null model (*bottom*), resembling no prediction of past and future locations, was included in the model comparison as baseline. Error bars denote ± s.e.m.; BIC, Bayesian Information Criterion (taking into account that the H0 model has fewer parameters). Two-tailed t test, ***p < 0.001, uncorrected for multiple comparisons.

separately (*Figure 3C*). Further, these results of greater activity for successor compared to predecessor activity also holds when comparing individual sequence locations without averaging (i.e., comparing non-stimulated location B when successor vs. predecessor, t(34) = 5.72, p = 2.02 × 10⁻⁶; and location C when successor vs. predecessor, t(34) = 3.13, p = 0.0035).

The activity decay toward distant future locations was formally tested by fitting an exponentially decaying factor gamma γ ∈ [0,1] to each participant's data. Here, values closer to 0 indicate a steeper decay and values closer to 1 indicate no decay. In line with our predictions, we found a group averaged decaying factor of γ=0.14 (±0.03 s.e.m.) that was statistically significantly different from 1 (non-parametric t test t(34) = −17.17, p = 2.54 × 10⁻¹⁸).

One might argue that participants with stronger predictions toward future locations would perform better at the behavioral detection task. However, no such correlation between individual V1 BOLD activity and task accuracy was found in an across-subject correlation analysis (see Materials and methods, spearman r = 0.05; p = 0.769).

## Successor-like representation in V1

Next, we sought to formally test how well the observed data fits the prediction of the SR, namely an exponential decay of states farther into the future.

For each subject, we fitted partial sequence trials with an SR model (*Figure 4A*), keeping the exponential decay parameter γ as a free parameter (see Materials and methods). In order to evaluate how well the SR model resembled the data, we then computed the error between the SR prediction and the actual data (lower values indicate a better model fit). For comparison, we additionally fitted a traditional pattern-completion co-occurrence (CO) model that predicts that events that occur together, will be reactivated together (*Figure 4B*). In contrast to the SR model, predictions of the CO model are non-directional, meaning that it predicts equal reactivation of both successor and predecessor locations. Furthermore, while the SR model predicts a temporal discounting toward future states, the CO model assumes no differential activity of reactivated states. In our implementation of the CO model, anticipatory activity was modulated by one multiplicative parameter $\omega$. Additionally, as a baseline model, we also evaluated a null model (H0) that assumes no predictive activity (i.e., no difference between successor and predecessor locations). Note that in order to be interpretable as a predictive representation, the best-fitting model should not only have the smallest error, but also differ significantly from the H0 model.

Our results show that anticipatory activity in V1 is best described by the predictions of the SR (*Figure 4C*; SR vs. CO t(34) = −2.29, p = 0.028). Additionally, both SR and CO describe the data better than the null model (SR vs. H0 t(34) = −8.25, p = 1.24 × 10⁻⁹; CO vs. H0: t test t(34) = −7.59, p = 8.22 × 10⁻⁹).

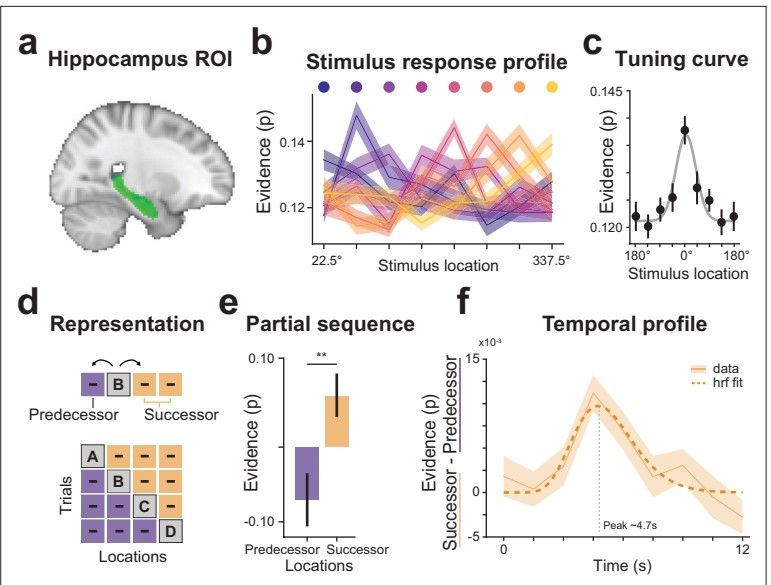

**Figure 5.** Hippocampus represents spatial locations and engages in future-directed predictions. (**a**) Hippocampus region of interest (green). (**b**) A pattern classifier was trained to distinguish between the eight stimulus locations during a perceptual localizer. Resulting stimulus-response profiles reveal that hippocampus distinguishes between individual stimulus locations. (**c**) Averaged (n=35) tuning profiles shifted to one location. (**d**) A classifier that was trained on the perceptual localizer was applied to partial sequence trials during the main task to probe whether hippocampal representations skew toward predecessor locations (purple), or successor locations (orange). (**e**) Classifier evidence, averaged across possible successor and predecessor locations, shows that hippocampus predominantly represents future (successor) stimulus locations over predecessor locations. (**f**) Since the hemodynamic properties of hippocampal functions are not well understood, the decoding analysis was additionally performed in a time-resolved manner and fitted with a canonical hemodynamic function to estimate the time to peak. The difference time-course (successor minus predecessor) showed a temporally distinct peak around 4.7 s indicating that the future-directed prediction occurs as transient response to the partial stimulus input and not as a sustained signal throughout the trial. Error bars denote ± s.e.m.; \*\*$p < 0.01$.

The online version of this article includes the following figure supplement(s) for figure 5:

**Figure supplement 1.** Successor-like representation of future sequence events in hippocampus.

## SR in hippocampus

The predictive neural representation in the form of a SR was originally postulated for the hippocampus (**Stachenfeld et al., 2017**). We therefore wanted to investigate whether the predictive representations that we observed in V1 were also present in the hippocampus. Note that while the hippocampal formation and nearby entorhinal cortex might feature a coarse representation of visual space (**Killian et al., 2012**; **Knapen, 2021**; **Nau et al., 2018b**; **Silson et al., 2020**), it does not feature the same fine-scale retinotopic organization present in V1 (**Dumoulin and Wandell, 2008**). Therefore, instead of focusing on univariate BOLD activity within certain hippocampal subregions, we focused on population activity patterns across the entire hippocampus using a decoding approach similar to previous studies (**Ekman et al., 2022**; **Kok and Turk-Browne, 2018**; **Kurth-Nelson et al., 2016**; **Russek et al., 2021**; **Schapiro et al., 2012**).

In keeping with the V1 analysis, we used the independent stimulus localizer to extract location-specific activity patterns in the hippocampus and then tested during partial sequence trials to what extent location-specific representations were reactivated. Specifically, we trained a pattern classifier to distinguish between the eight dot locations within the localizer. Before applying the trained classifier to omission trials of the main task (see Materials and methods), we confirmed that cross-validated decoding accuracies within the localizer were above chance-level to ensure that the hippocampal pattern shows a reliable representation of space.

Within localizer decoding accuracy results confirmed that hippocampus has a coarse representation of the eight stimulus locations (**Figure 5B**) within the localizer (two-sided one-sample t test; t(34)

= 3.28, p = 0.002; cross-validated accuracy = 15 ± 3.6%, mean ± s.d.; see Materials and methods). Notably, compared to V1 (*Figure 2A*), within localizer accuracy was relatively low and as a consequence tuning curves in hippocampus appeared less sharp (*Figure 5C*). In order to maximize sensitivity for the hippocampus, we averaged classification evidence across successor and predecessor locations. Non-averaged results can be found in *Figure 5—figure supplement 1*.

Applying the trained classifier to partial sequence trials of the main task, we asked whether hippocampus would preferentially reactivate successor or predecessor locations (*Figure 5D*). To answer this question, we first subtracted the probabilistic classifier evidence for the control locations from the classifier evidence of the sequence locations. Consequently, values greater than 0 reflect evidence for the reactivation of sequence representations, while values smaller than 0 reflect a relative suppression of sequence locations. After that we averaged the evidence across all successor and predecessor locations, respectively, and tested for differences across participants. Our results reveal that hippocampus representations were preferentially biased toward successor locations (*Figure 5E*; paired-sample t test, t(34) = 2.74, p = 0.009), mirroring the results found in V1.

Finally, in order to better understand the temporal dynamics of the anticipatory representations in hippocampus, we repeated the decoding analysis in a time-resolved manner. We reasoned that if reactivations of future sequence locations were triggered by the brief presentations of partial sequence dots, the evidence time-course should follow a transient response profile. Alternatively, if hippocampus were to signal a constant bias toward future sequence locations, the evidence time-course should be unrelated to the stimulus onset and show a sustained temporal profile.

Results of the evidence difference time-course clearly show a transient response peaking approximately 4.7 s post-stimulus onset (*Figure 5F*) indicating that hippocampal predictions were triggered by the partial sequence dot. Note that the decoding time-course reflects the evidence for successor locations versus predecessor locations independent of the bottom-up stimulus. The transient decoding profile can therefore not simply reflect the onset of a given trial.

In order to probe the relationship between hippocampus and V1 successor reactivations, we performed an across-subject analysis, correlating V1 BOLD activity, averaged across all successor locations, with hippocampus classifier evidence, averaged across all successor locations. No significant relationship was observed (spearman correlation, r = –0.08, p = 0.668).

One could ask whether our findings are specific to V1 and hippocampus, or widespread throughout the brain. In order to answer this question, we repeated the analysis for low-level visual area V2. In contrast to V1, no predictive effects were found in area V2. V2 BOLD activity was not enhanced at the non-stimulated successor locations compared to the non-stimulated predecessor locations (averaged across all partial trials and sequence locations; t(34) = 1.41, p = 0.168).

## Hippocampal codes preserve spatiotemporal tuning

In contrast to V1, hippocampal representations are not inherently retinotopic and feature only a coarse representation of visual space (*Knapen, 2021*; *Nau et al., 2018b*; *Silson et al., 2021*). Instead, hippocampal place cells provide a detailed representation of the allocentric position in an environment. However, more recently, the intriguing picture emerged that hippocampus also contributes to a more general organization of information by representing non-spatial aspects of experience in a map-like way (*Constantinescu et al., 2016*; *Garvert et al., 2017*; *Stachenfeld et al., 2017*), similar to the representation of space (*Aronov et al., 2017*).

Inspired by these recent observations, we asked what the underlying properties of the reported hippocampus representations were. Given that we successfully trained a classifier based on eight spatial locations, it might seem obvious to conclude that the underlying code for these representations is purely spatial (retinotopic) as well. This is however not necessarily the case, given that the localizer was shown after the main task and might therefore reflect persistent predictive representations. Instead, robust discrimination of sequence locations could theoretically also be based on coding of temporal properties of the sequence. Indeed, *Deuker et al., 2016* have recently shown that hippocampus representations can reflect in principle both spatial and temporal aspects. In our case, a temporal coding mechanism could represent stimulus locations not based on proximity in space, but rather by proximity in time.

In order to address this question, we conducted a detailed analysis of the coactivation pattern in the stimulus localizer (*Figure 6A*). Note that the localizer was shown at the end of the study, allowing

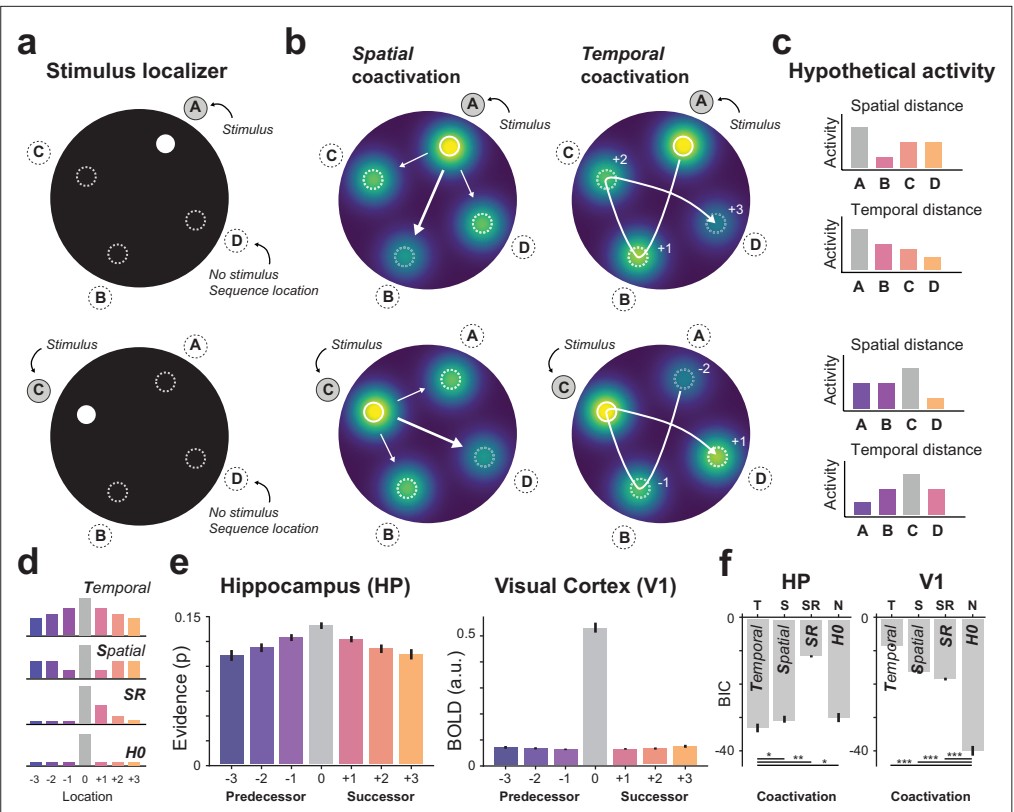

**Figure 6.** Stimulus localizer reveals complementary coactivation (tuning) properties in hippocampus and V1.
(**a**) Schematic of the localizer trial with the stimulated location 'A' and the non-stimulated locations (B, C, D, dashed circle) that were part of the sequence in the main task preceding the localizer. (**b**) Illustration of coactivation (tuning) of sequence locations based on spatial (Euclidean) distance from the stimulated location (*left*) and temporal distance in sequence space (*right*). Note how sequence locations A and B are far apart in the spatial (Euclidean) domain, but close in terms of temporal distance in sequence space. (**c**) Hypothetical activation pattern for representational tuning of spatial distance and temporal distance for illustration shown in (**b**). (**d**) Illustration of tuning pattern averaged across all localizer conditions for temporal tuning (*top*), spatial tuning (*middle*), Successor Representation (SR, middle), and no coactivation (H0, *bottom*). For visualization purposes the x-axis is sorted by time for all three tuning patterns. (**e**) Classifier evidence for current, future, and past locations for hippocampus (*left*) and V1 (*right*). (**f**) Comparing model errors (i.e., lower is better) show that hippocampal representations were best described by temporal coactivation (*left*), while V1 (*right*) was best described by spatial coactivation and the absence of coactivation (H0) of sequence locations. Error bars denote ± s.e.m.; two-tailed t test, ***p < 0.001; **p < 0.01; *p < 0.05, uncorrected for multiple comparisons; *BIC*, Bayesian Information Criterion (taking into account that the H0 model has fewer parameters).

us to test whether learned associations persisted even after the full sequence was not relevant anymore. Here, coactivations were defined as activation of non-stimulated locations. For instance, when presenting stimulus A, locations B-C-D might become activated as well. In general, such coactivations are often attributed to noise or ambivalent responses driven by overlapping receptive fields. However, in this case, we made use of the coactivation pattern to draw inferences about the learned persistent representations.

Specifically, for a representation based on spatial coactivation (tuning) one would expect a coactivation of nearby spatial locations. In other words, the coactivation of non-stimulated locations should be modulated by the spatial (Euclidean) distance to the stimulated location (*Figure 6B*). This spatial tuning pattern is typically seen in early visual areas with overlapping receptive fields and is also visually present in our V1 results (*Figure 2A*). Alternatively, for a representation based on temporal coactivation (tuning) one would expect a coactivation of nearby locations in sequence space (*Figure 6C*).

Thus, spatial and temporal tuning codes lead to different coactivation patterns (*Figure 6D*) that can be disentangled with our stimulus paradigm. For instance, sequence locations A and B were far apart

in the spatial (Euclidean) domain, but close in the temporal domain (distance in sequence space). Conversely, locations A and D are close in terms of spatial distance, but far apart in terms of temporal distance (*Figure 6B*). Note, while spatial tuning is in principle independent of any task-specific experience, temporal tuning on the other hand requires exposure to a sequential structure and can therefore only occur for the four dots that were part of the sequence. For this reason, we restricted the coactivation (tuning) analysis to the four dot locations that were part of the sequence.

For each participant, individual localizer data were fitted by a spatial coactivation model, a temporal coactivation model, an SR model, and a no-coactivation (H0) control model (*Figure 6D*). The latter was included as a low-level baseline control. Visual inspection of the group averaged localizer coactivation pattern revealed a clear temporal tuning pattern in hippocampus but not in V1 (*Figure 6E*). These results were confirmed by a formal model comparison (*Figure 6F*, *Hippocampus*: two-sided t test, Temporal vs. Spatial $t(34) = -2.36$, $p = 0.024$; Temporal vs. SR $t(34) = -3.24$, $p = 0.003$; Temporal vs. H0 $t(34) = -19.27$, $p = 7.12 \times 10^{-20}$; V1: H0 vs. Temporal $t(34) = -22.09$, $p = 9.49 \times 10^{-22}$; H0 vs. Spatial $t(34) = -14.26$, $p = 6.52 \times 10^{-16}$; H0 vs. SR $t(34) = -17.15$, $p = 2.61 \times 10^{-18}$).

## Discussion

Uncovering the computations that drive human prediction and planning is a central aspect when it comes to understanding human cognition. What are the general coding mechanisms that allow to utilize knowledge of the environment to make inferences and generalizations about future events? In this study, we sought to answer the question whether the map-like SR that has been posited for the hippocampus (*Mehta et al., 2000*; *Stachenfeld et al., 2017*) may also explain the shape of anticipatory activity in visual cortex (V1).

There is an extensive body of literature that shows how expectations elicit anticipatory activity in early visual cortices (*de Lange et al., 2018*; *Hindy et al., 2016*; *Kok et al., 2012*). For instance, we have previously shown that flashing an individual dot of a simple, linear sequence triggers an activity wave in V1 that resembles the full stimulus sequence (*Ekman et al., 2017*; *Ekman et al., 2022*), akin to replay of place field activity during spatial navigation (*Foster and Wilson, 2006*; *Gupta et al., 2010*). However, what remains unknown is whether these sensory replay traces are guided by a generative model that represents the relational structure of the stimulus sequence, akin to a predictive map. Alternatively, anticipatory activity traces could simply reflect the association between different stimuli, based on their CO, without the added complexity of any temporal relational structure. The latter explanation appears plausible, given that predictive representations in early visual cortex are generally time critical and operate in parallel to a constant stream of new sensory input, which arguably requires efficient processing and in turn limits the complexity of such representations.

In fact, we previously speculated that cue-triggered reactivation of simple sequences might be driven by an automatic pattern completion-like mechanism that reactivates all associated items based on partial input (*Ekman et al., 2017*). This idea is in line with the finding that predictive representations in V1 correlated with pattern completion-like activity in the hippocampus (*Hindy et al., 2016*; *Kok and Turk-Browne, 2018*) that might be driving V1 activity (*Finnie et al., 2021*; *Ji and Wilson, 2007*).

Our current findings directly challenge this interpretation and instead point to a predictive representation of expected, temporally discounted, future states. We accomplished this by using a paradigm in which one visual event (e.g., the presentation of one dot) was framed as one state in a directed transition matrix with a fixed relational structure. The SR hypothesis makes two testable predictions, namely that population activity represents future states over predecessor states, and that future state representations are temporally discounted, such that events in the close future are more prominently represented compared to events in the distant future. Using a paradigm in which we occasionally presented only single items of the full sequence, allowed us to investigate V1 activity at omitted sequence locations.

Confirming the SR predictions, V1 activity at the successor locations was enhanced compared to activity at the predecessor locations, indicating a representation skewed toward future locations and away from the past. Notably, this relative difference was not only due to an enhancement of successor states, but our results also showed a decrease of activity at the predecessor states (compared to baseline). This suppression of predecessor states might seem surprising at first given that SR postulates the mere absence of predecessor activity (*Momennejad, 2020*; *Stachenfeld et al., 2017*). We speculate

that the observed decrease at the predecessor states might constitute a functional separation mechanism between predecessor and successor states, strengthening the future-directed representation of the sequence by selectively decreasing representations of the unexpected predecessor states.

One aspect that sets our study apart is that the viewing of the visual sequence does not require any predictive planning of the participant to evaluate different future outcomes. In contrast, related studies reporting neuronal evidence for SR-like representations in hippocampus and PFC (*Barron et al., 2020*; *Brunec and Momennejad, 2022*) and occipital cortex (*Schwartenbeck et al., 2021*) have used paradigms in which participants were actively engaged in prospective planning and choice evaluation. Given the relatively passive nature of our task, one might therefore wonder whether it is expected to find any map-like activity at all. However, in this context, it is important to stress that the SR, unlike other model-based algorithms, does not depend on choice-dependent reward to build its transitional task structure (*Momennejad et al., 2017*; *Stachenfeld et al., 2017*) and therefore might not depend on participants' active engagement. Furthermore, *Russek et al., 2021* have recently used a paradigm in which subjects were passively exposed to transitions between visual states and reported evidence for SR-like representations in the absence of active choices in line with the results of the present study. Further supporting this notion, we have previously shown that anticipatory sequence activity occurred even after subjects' attention was diverted from the sequence to a demanding task at fixation (*Ekman et al., 2017*), rendering the sequence task irrelevant. Taken together, these observations indicate that SR-like representations are not limited to situations that require active planning, or multiple-choice evaluations but may rather be formed automatically and incidentally, as has been shown repeatedly in the domain of statistical learning (*Fiser and Aslin, 2002*; *Turk-Browne et al., 2005*).

While we have interpreted the neural activity patterns in the light of the SR, it is strictly speaking not possible to distinguish between model-based (MB) and SR algorithms within the context of our design. The key distinction between them is that SR caches a predictive map of states that the agent expects to visit in the future, whereas MB algorithms store a full model of the world and compute trajectories at the decision time (*Gershman, 2018*; *Momennejad et al., 2017*). Therefore, both predict a temporally discounted activation of successor states. It should be noted however that MB comes at a higher computational cost, and is more intensive both in terms of time and working memory resources. The activation of successor states that we observed, on the other hand, occurred in the absence of a decision-making process (i.e., participants did not perform any task on the trials where a single dot was presented). Also, importantly, we previously observed that this activation pattern was not dependent on the task, and was equally present when attentional resources were strongly drawn away from the stimuli (*Ekman et al., 2017*). These observations may be more readily in line with the automatic (cached) activation of successor states that is embodied by SR, rather than the effortful iterative calculation of successor states that is the hallmark of MB. One future possibility to disentangle SR and MB algorithms could be to probe how well each model adapts to changes in the dot sequence structure. It has previously been shown, that compared to MB, the flexibility of the SR is somewhat limited to reflect changes in the transitional structure, because it requires the entire SR to be relearned (*Momennejad et al., 2018*).

The hippocampal formation can acquire arbitrary relationships between objects (*Aronov et al., 2017*; *Backus et al., 2016*; *Behrens et al., 2018*; *Constantinescu et al., 2016*; *Garvert et al., 2017*) beyond geometric location in space (*O'Keefe and Nadel, 1978*). While our main focus in the current study was on V1 representations, we also wanted to test to what extent hippocampus showed a similar SR-like representation of visual sequences. Previous fMRI studies investigating hippocampal representations have mainly focused on either navigation in a spatial (*Brunec and Momennejad, 2022*; *Deuker et al., 2016*) or non-spatial task (*Garvert et al., 2017*; *Schapiro et al., 2013*; *Schuck and Niv, 2019*) in which participants explore a relatively complex task space. It was recently shown that hippocampus has a rudimentary representation of visual space (*Knapen, 2021*; *Silson et al., 2021*), but it was not clear whether hippocampus would also engage in the representation of a comparably simple, low-level visual sequence presented in our paradigm.

Our results confirmed that hippocampus representations resemble an SR-like predictive map, favoring future over past sequence locations. This result highlights the compelling conceptual parallels between mnemonic expectations in hippocampus (*Hindy et al., 2016*) and its perceptual manifestation in sensory cortex. On a conceptual level, navigation (in memory and space) and processing visual events both involve abstraction of the relational structure between events to enable forward planning

and predictions. Similar to navigational space, visual space can be represented in terms of its relational structure-like direction and distance, and it has been suggested that similar mechanisms might underlie spatial and non-spatial representations (*Nau et al., 2018a*), especially if there is sequential structure present (*Finnie et al., 2021*). Supporting this notion, recently, a conceptual link between representations for visual understanding and spatial navigation has been proposed that suggests a common underlying map-like representation of visual and navigational task structure (*Schwartenbeck et al., 2021*).

These conceptual links, as well as anatomical (*Felleman and Van Essen, 1991*; *Huang et al., 2021*) and functional connections between the hippocampus and visual cortex (*Bosch et al., 2014*; *Hindy et al., 2016*; *Ji and Wilson, 2007*; *Kok and Turk-Browne, 2018*; *Lee et al., 2012*; *Nau et al., 2018a*) raise the question whether hippocampal representations are independent from V1, or whether V1 is instead receiving the predictions as a feedback signal from hippocampus. Supporting the idea of functional feedback, *Finnie et al., 2021* recently showed that V1 predictions were heavily impaired after hippocampus damage. However, contrary to this notion, spatiotemporal sequence predictions have also been shown to occur locally within V1 without the need for top-down predictions (*Gavornik and Bear, 2014*; *Xu et al., 2012*). Our study showed no functional relationship between sequence prediction in V1 and hippocampus. However, our experimental paradigm was not primarily designed to address this question, as it does not exclude the possibility that an apparent coordination might be driven by other factors like attentional fluctuations across participants. Further, V1-hippocampus coordination might exist on a trial-by-trial level, which does not necessarily transfer to statistical comparisons across participants. Future experiments, using more than one stimulus sequence could potentially address this question by comparing evidence of sequence-specific representations in both areas within participants.

It is notable that while hippocampal and visual representations appear similar with respect to their SR-like representation, they also show qualitative differences with respect to their underlying coding properties. V1 representations of individual sequence items resembled a coding based on spatial tuning. Hippocampus on the other hand represented relevant items predominantly in terms of their temporal distance within the sequence, suggesting that representations capitulate on the transitionally structure of the visual sequence. These results align with previous reports that hippocampus can learn to represent temporal sequence structure (*Thavabalasingam et al., 2018*; *Thavabalasingam et al., 2019*) and temporal proximity in a spatial navigation task (*Deuker et al., 2016*; *Howard et al., 2014*), but to the best of our knowledge, constitute the first reports of coding temporal distance of a visual sequence.

Furthermore, hippocampus predictive codes were found to persist after the sequence task and coactivation of related sequence locations was still present during the stimulus localizer, potentially indicating that hippocampus representations reflect a more stable code operating on a longer timescale. V1 representations on the other hand did not persist throughout the stimulus localizer and reverted back to representing individual spatial locations without coactivation of related sequence locations, further highlighting another qualitative difference between V1 and hippocampus coding. According to the SR, it is expected that sequence predictions will change once the regularities of the environment change. The absence of SR-like pattern in V1 during the functional localizer is therefore not at odds with our results from the main task, but rather indicative of a dynamic updating of the generative model. Taken these qualitative differences together, it is reasonable to speculate that predictive activity in V1 does not merely reflect top-down feedback from hippocampus, but instead that SR-like representations in V1 are somewhat independent, and potentially complementary, to SR-like representations found in the hippocampus.

In conclusion, our data show that anticipatory activity in early visual cortex and hippocampus is guided by a generative model that represents the relational structure of the visual world, akin to a predictive map. Our results suggest that the observed SR-like representation underlying visual predictions can provide a sophisticated state space representation that enables flexible generalization from partial input to future sequence locations, while also being efficient enough to provide rapid visual computations.

## Materials and methods
### Preregistration
The experimental design, data analyses, and hypotheses were all preregistered at Open Science Framework (https://osf.io/f8dv9/) prior to data collection.

## Participants

Thirty-seven right-handed subjects participated in the fMRI study. Two participants were excluded based on predetermined performance and motion criteria during scanning (error rate/relative motion three standard deviations above the group mean). The final sample included 35 subjects (20 females, mean age = 27 years). Target sample size was decided prior to data collection based on a power analysis (two-sided paired t test, power = 80%, Cohen's d ≥ 0.5 and α = 0.05). Participants gave written informed consent in accordance with the institutional guidelines of the local ethical committee (CMO region Arnhem-Nijmegen, The Netherlands) and received monetary compensation for their participation. All participants had normal or corrected-to-normal visual acuity.

## Stimuli

Participants viewed a sequence of four white dots on a black background. Dot locations were sampled from eight possible locations (*Figure 1B*). The center of each dot location was 7 degrees visual angle (dva) away from the central white fixation cross (0.5 dva) and the locations were equally spaced around the center (distance in polar angle from the vertical line: 22.5°, 67.5°, 112.5°, 157.5°, 202.5°, 247.5°, 292.5°, and 337.5°, see *Figure 1B*). The dots had a diameter of 1.2 dva. Stimulus sequences were shown on an MRI safe LCD screen (BOLDscreen 32, 1920 × 1080 pixel resolution, 60 Hz refresh rate). Participants were positioned 134 cm away from the screen and viewed the stimuli via a mirror on top of the head coil.

During full sequence trials, each dot was shown for 100 ms with an interstimulus interval (ISI) of 17 ms, resulting in a total sequence duration of 451 ms. For 52 out of 128 full sequence trials, the onset of the last dot was delayed with an ISI of 170 ms (instead of 17 ms). Participants were instructed to detect and report these delayed sequence presentations via a button press with their right index finger.

Sequences were constructed such that each of the eight locations served as a starting location for one possible sequence. Further, each quadrant was stimulated once, which also excluded the possibility that neighboring dots were part of the same sequence. This constraint was chosen to minimize the potential spreading of activity from one location to neighboring sequence locations. Specifically, the second dot was always presented opposite of the starting location (180° clockwise from the start). The third dot was shown 90° clockwise from the second location and the last dot was on the opposite side of the third location. These constraints also served to decouple spatial and temporal distance within the sequence. With these constraints, there were eight possible visual sequences that were randomly assigned and counterbalanced in frequency across subjects. Dots that were part of the sequence are labeled as sequence dots A-B-C-D. While the remaining four dots at locations that were not part of the sequence are referred to as 'control dots'.

Note that because within each dot sequence, temporal order and spatial distance were not perfectly decorrelated (e.g., the second sequence dot was always farthest apart from the starting dot), it is not possible to estimate the combined influence of the SR model and the spatial coactivation model on the observed BOLD activity.

## Experimental design

The experiment lasted a total of 2 hr and consisted of three blocks (i) learning, (ii) main task, and (iii) a stimulus localizer. During the learning part, participants were familiarized with one of the eight sequences. The full sequence, consisting of four successively presented dots A-B-C-D, was shown 352 times outside and 160 times inside the scanner. In order to maintain participants' attention during the learning part, there was a delay detection task on 50% of the trials. Participants were instructed to detect a timing delay of the last dot for which they had 1 s to respond. After every 30 trials, participants were shown their aggregated detection accuracy. During the initial learning phase outside the scanner, participants received additional feedback after each trial on whether their response was correct or incorrect through changes in the color of the fixation cross (green for correct and red for incorrect answers). No trial-wise feedback was given inside the fMRI. Participants were instructed to maintain fixation throughout the experiment and eye movements were measured with an Eyelink 1000 eye-tracker system (SR Research, Ontario, Canada; 1000 Hz sampling rate).

The main task consisted of three runs of equal duration (about 13 min). There were 192 trials per run and 576 trials in total. Trials were separated by a variable inter-trial interval (ITI) with a duration drawn from a truncated exponential distribution with a minimum of 2 s, maximum of 10.9 s, and mean

of 3.72 s. The variable ITI ensured that the experimental paradigm had no temporal structure that participants could learn to expect the onset of a trial. This allowed us to focus in the present study on the learning and representation of structural knowledge, independent of any temporal expectation effects.

To probe activity replay, we introduced partial sequence trials where only one of the four dots was shown for 100 ms, instead of the full sequence. Visually, there was no difference between the ITI and the part of the partial sequence trials where the dots were omitted, both showed a fixation cross at the center of the screen. During each run, two thirds of the 192 trials were full sequence trials (128 trials) and one third of the trials were partial sequence trials (64 trials). Trial order was pseudo-randomized with the constraint that partial sequence trials were always followed and preceded by a full sequence trial, excluding the possibility of partial sequence trial repetitions. The pseudo-randomization (perfect counterbalancing was numerically not possible with the set number of trial types and repetitions), rules out the possibility of systematic order effects. There was a task on ~40% of the full sequence trials (156/384 trials). At the end of each run, participants received feedback on their performance.

After the main task, we ran a functional localizer (~16 min) where each dot was flashed at 2 Hz for 13.5 s in a pseudo-randomized order, followed by 15 s rest period. In total, each dot location was presented eight times and each of the eight dots followed once immediately after the rest period. Participants performed a letter stream task at fixation where they had to detect target letters ('X' and 'Z') in a stream of non-target letters ('A', 'T', 'N', 'U', 'V', 'Y', 'H', and 'R'). The target probability was 10%. Each letter was presented for 500 ms.

For a small subset of N = 7 participants, after the localizer, we additionally presented moving bar stimuli, in order to map the pRFs of voxels in early visual cortex. During these runs, bars containing full-contrast flickering checkerboards (2 Hz) moved across the screen in a circular aperture with a diameter of 20°. The bars moved in eight different directions (four cardinal and four diagonal directions) in 20 steps of 1°. Four blank fixation screens (10.8 s) were inserted after each of the cardinally moving bars. Throughout each run (5.76 min), a colored fixation dot was presented in the center of the screen, changing color (red to green and green to red) at random time points. Participants' task was to press a button whenever this color change occurred. Participants performed four identical runs of this task.

## MRI acquisition

Functional and anatomical MRI data were acquired on a 3 T PrismaFit scanner (Siemens AG, Healthcare Sector, Erlangen, Germany) using a 32-channel head coil. The protocol included a T1-weighted anatomical scan and five functional runs. The anatomical scan was acquired with a Magnetization Prepared Rapid Acquisition Gradient Echo sequence (MP-RAGE; TR = 2300 ms, TI = 1100 ms, TE = 3 ms, flip angle = 8°, $1 \times 1 \times 1$ mm$^3$ isotropic). To acquire the functional images, we used a T2*-weighted multiband 4 (*Moeller et al., 2010*) sequence (TR = 1500 ms, TE = 39 ms, flip angle = 75°, $2 \times 2 \times 2$ mm$^3$, 68 slices). The five functional runs comprised of one learning run, three main task runs, and one localizer run. For two subjects only two main task runs were acquired because of time constraints. Seven participants participated in a previous study in which they completed four runs of pRF mapping.

## fMRI preprocessing

MRI data were preprocessed using FSL (version 6.00; FMRIB Software Library) (*Smith et al., 2004*). We applied brain extraction using BET, motion correction using MCFLIRT, temporal high-pass filtering (100 s) and spatial smoothing (Gaussian kernel, FWHM = 5 mm). All analyses were carried out in native subject space. The first three volumes of each run were discarded to allow for signal stabilization. Registration of the functional images to the anatomical image was performed with FLIRT boundary-based registration. The anatomical image was registered to the MNI152 T1 2 mm standard space template (linear registration, 12 degrees of freedom).

## ROI selection

V1 and hippocampus ROIs were determined using the automatic cortical parcellation provided by Freesurfer (*Fischl, 2012*) based on individual T1 images. Anatomical V1 and hippocampus masks were then transformed into native space using linear transformation. For V1, we used a preregistered voxel selection method to determine V1 subpopulations that are most responsive to individual stimulus locations.

First, the localizer data were fitted with a voxel-wise general linear model (GLM) using FSL FEAT (*Smith et al., 2004*) with the following regressors: 8 regressors of interest for stimulation of each of the locations (duration = 13.5 s), 1 regressor for the instructions and the end-of-block screen (duration = 4.5 and 15 s, respectively) as well as the 24 FSL motion regressors.

Second, for each location, we calculated the GLM contrast by comparing one location to all other locations and selected the 25 most selective voxels (highest z-values). Third, we removed voxels from the selection that were selective for multiple dot locations. Finally, we determined the lowest number of selective voxels per region and removed the least active voxels from all other locations until all V1 subpopulations had the exact same number of selected voxels per location. This procedure was chosen to rule out the possibility that potential activity differences across locations could be attributed to different number of voxels per region. Across subjects, we selected on average 22.05 voxels (SD = 2.88) per location.

## V1 BOLD amplitude modulation

A GLM for the main task was created with the following regressors: 8 regressors for each single dot trial (4 sequence dots and 4 control dots), 1 regressor for the full sequence trial, 1 regressor of no interest to model the instructions and the feedback at the end of a run and 24 motion regressors (6 standard and 18 extended FSL motion parameters, i.e., the derivatives of the standard motion parameters, the squares of standard motion parameters, and the squares of the derivatives). Note that the control dot trials in the main task were modeled in the GLM, but treated as regressors of no interest. The model was convolved with a single gamma hemodynamic response function. Nine contrasts were set up that tested which voxels were more responsive to presentation of a single dot (eight contrasts, one for each dot) or the full sequence (one contrast) compared to baseline. The GLM was fit to each run separately and resulting beta estimates were averaged across runs for each participant. In order to obtain an estimate of stimulus-specific activity (*Figure 2B*), we averaged the activity at the four control ROIs and subtracted it from the activity at the sequence ROIs.

## Correlation with behavior

In order to relate SR representations to behavior, we first calculated individual V1 BOLD differences for all successor versus all predecessor locations to get an estimate for how much participant's predictions were skewed toward future locations. We then correlated these values with behavioral accuracy across subjects using Spearman correlation.

## V1 model comparison

For each participant, V1 BOLD activity from the partial trials was fitted with three models, SR, coactivation (CO), and a null-model (H0). The resulting root mean square error (RSME, lower values = better fit) between model fit and observed data was then tested across participants for significance using paired-sample t tests to address the question whether one model prediction describes the underlying data better than competing models.

The model prediction of the SR is based on the task structure, formalized in a transition matrix $T$ of the sequence A-B-C-D (*Figure 4*). The SR matrix $M$ is then calculated as:

$$M = \left(I - \gamma T\right)^{-1}$$

where I is the identity matrix and $\gamma \in \left[0, 1\right]$ is the discount factor or predictive horizon. During model fitting, γ was a free parameter, meaning that instead of using a fixed value, individual γ values were determined for each participant. Here, larger values of γ result in a smaller exponential decay of future states. The model prediction of the CO model is based on the CO of events. In contrast to SR, the task structure is non-directed and off-diagonal values in the CO model are constant and modulated in amplitude by a free multiplicative parameter $\omega$. The H0 (null) model serves as a baseline that assumes no off-diagonal (predictive) activity. In order to be interpretable any winning model should outperform the H0 model. The diagonal values in all three models reflect the bottom-up stimulation induced by the single dot of the partial trials.

## Hippocampal decoding

The decoding analysis was performed with scikit-learn (*Pedregosa et al., 2011*). Individual voxel time courses were low-pass filtered using a Savitzky-Golay filter with a window length of 5 TRs and polynomial order of 3 (*Savitzky and Golay, 1964*) and normalized to z-scores. Volumes for individual localizer trials were averaged between 3 and 13.5 s to capture only stimulus-related BOLD activity. A logistic regression classifier (default values, L2 regularization; C = 1) was trained to distinguish between eight stimulus locations during the independent localizer run. Before applying the trained classifier to the main task, we confirmed that the classifier was indeed able to distinguish between stimulus locations within the localizer. To this end, we performed a leave-one-out cross-validation and tested the decoding accuracy against chance level (1/8 = 12.5 %) across subjects using a one-sample t test. In addition to a binary classifier output for each class, we also looked at the probabilistic output. For each sample in the localizer test set, we obtained eight probability values, one for each class. We refer to the classifier probability as classifier evidence, as the probability reflects the evidence that a particular class is represented. For each participant, probability values were averaged across trials to obtain location-specific response profiles.

Next, we trained the classifier on all localizer trials and applied it to individual trials of the main task. Volumes for individual main task trials were averaged between 3 and 6 s to capture only stimulus-related BOLD activity. Note that the main task was an event-related design with shorter trial durations compared to the block-design localizer with 13.5 s stimulation periods; hence, the different averaging windows of 3–13.5 and 3–6 s. Similar to the BOLD analysis in V1, for each partial sequence trial in the main task, we averaged the classifier evidence for the four control locations and subtracted it from the evidence of the sequence locations. We then averaged the classifier evidence for all predecessor and successor locations, respectively, and compared the evidence across subjects with a paired-sample t test.

Finally, in order to rule out that the chosen time window had any influence on the results, we repeated the decoding analysis in a time-resolved manner, repeating the steps above for each volume from 0 to 13.5 s separately. Fitting a standard hemodynamic response function (hrf) revealed a transient decoding evidence peak at around 4.7 s.

## Hippocampus and V1 tuning

The tuning analysis investigates coactivation pattern during the localizer and focuses on the four locations that were part of the stimulus sequence in the preceding main task. Classifier evidence values within the localizer were averaged and sorted to reveal potential coactivation (tuning) pattern of sequence locations. Three tuning patterns were considered and tested: (i) temporal tuning, assuming a linear decay from the currently presented stimulus toward location that where farther in the past and future (two free parameters, slope, and intercept), (ii) spatial tuning, assuming a linear decay from the current stimulus toward other stimulus locations modulated by spatial distance (two free parameters, slope, and intercept), and (iii) a baseline no-coactivation pattern. Note that the latter model was considered because V1 tuning curves were rather sharp with little activity spread to immediate neighboring locations (5.4° apart; *Figure 2A*) and locations in the current analysis were 9.9° apart. For each participant, aggregated classifier evidence was fitted using three tuning patterns and resulting errors were compared across subjects to determine the best-fitting pattern. Fitting was performed using the curve_fit function in SciPy 1.6.2 (*Virtanen et al., 2020*).

## pRF estimation and reconstruction

pRF data were available for seven participants from a previous study (*Ekman et al., 2022*) and were used to validate visually that the voxel selection based on the functional localizer selected voxel that corresponds to the stimulated location is visual space (*Figure 2*). Data from the moving bar runs were used to estimate the pRF of each voxel in the functional volumes using MrVista (http://white.stanford.edu/software). In this analysis, a predicted BOLD signal is calculated from the known stimulus parameters and a model of the underlying neuronal population. The model of the neuronal population consisted of a two-dimensional Gaussian pRF, with parameters $x_0$, $y_0$, and $\sigma_0$, where $x_0$ and $y_0$ are the coordinates of the center of the receptive field, and $\sigma_0$ indicates its spread (standard deviation), or size. All parameters were stimulus-referred, and their units were degrees of visual angle. These parameters were adjusted to obtain the best possible fit of the predicted to the actual BOLD signal.

This method has been shown to produce pRF size estimates that agree well with electrophysiological receptive field measurements in monkey and human visual cortex (*Klink et al., 2021*). For details of this procedure, see *Dumoulin and Wandell, 2008*; *Kay et al., 2015*. Once estimated, x0 and y0 were converted to eccentricity and polar-angle measures and co-registered with the functional images using linear transformation.

For the pRF-based stimulus reconstruction, for each participant, we first limited the pRF data to voxel that were selected based on the functional localizer (25 voxels per stimulus location, 200 voxels in total). This selection step would allow us to visually inspect whether the voxel selection accurately selected voxel corresponding to the respective stimulus location. Second, every voxel is described as a 2D Gaussian with parameters x0, y0, and s0 from the pRF estimation. The 2D Gaussians for each voxel, represented by a pixel × pixel image, were scaled based on the percent signal change obtained from the functional localizer GLM, and consecutively summed over voxels to create one 2D representation of the reconstructed stimulus. This procedure was repeated separately for all eight stimulus locations. Finally, for visualization purpose, the eight individual localizer conditions were rotated to one stimulus location (22.5°) and averaged across stimulus locations and participants.

# Additional information

### Competing interests
Floris P de Lange: Senior editor, *eLife*. The other authors declare that no competing interests exist.

### Funding

| Funder | Grant reference number | Author |
| --- | --- | --- |
| Nederlandse Organisatie voor Wetenschappelijk Onderzoek | Veni Grant No. 016. Veni.195.435 | Matthias Ekman |
| HORIZON EUROPE European Research Council | ERC Consolidator Grant 101000942 "Surprise" | Floris P de Lange |

The funders had no role in study design, data collection and interpretation, or the decision to submit the work for publication.

### Author contributions
Matthias Ekman, Conceptualization, Resources, Data curation, Software, Formal analysis, Supervision, Funding acquisition, Investigation, Visualization, Methodology, Writing - original draft, Project administration, Writing - review and editing; Sarah Kusch, Conceptualization, Formal analysis, Methodology, Project administration, Writing - review and editing; Floris P de Lange, Conceptualization, Supervision, Funding acquisition, Methodology, Writing - review and editing

### Author ORCIDs
Matthias Ekman ![ORCID] http://orcid.org/0000-0003-1254-1392
Floris P de Lange ![ORCID] http://orcid.org/0000-0002-6730-1452

### Ethics
Human subjects: The study followed institutional guidelines of the local ethics committee (CMO region Arnhem-Nijmegen, The Netherlands; Research Protocol "Imaging Human Cognition", NL45659.091.14), including informed consent of all participants.

### Decision letter and Author response
Decision letter https://doi.org/10.7554/eLife.78904.sa1
Author response https://doi.org/10.7554/eLife.78904.sa2

## Additional files

### Supplementary files
• MDAR checklist

### Data availability
All data and code used for stimulus presentation and analysis are available on the Donders Repository (https://doi.org/10.34973/bsy6-9h29).

The following dataset was generated:

| Author(s) | Year | Dataset title | Dataset URL | Database and Identifier |
|---|---|---|---|---|
| Ekman M, Kusch S, de Lange FP | 2023 | Successor-like representation guides the prediction of future events in human visual cortex and hippocampus | https://doi.org/10.34973/bsy6-9h29 | Donders Repository, 10.34973/bsy6-9h29 |

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
