## [Editor Report]

In this paper, Ekman and colleagues present compelling fMRI evidence from a visual sequence task that both the early visual cortex (V1) and the hippocampus represent perceptual sequences in the form of a predictive "successor" representation, where the current state is represented in terms of its future (successor) states in a temporally discounted fashion. In both brain structures, there was evidence for upcoming, but not preceding steps in the sequence, and these results were found only in the temporal but not spatial domain. This study offers the fundamental suggestion that both the hippocampus and V1 represent temporally structured information in a predictive, future-oriented manner.

---

## [Decision Letter]

**Decision letter after peer review:**

Thank you for submitting your article "Successor-like representation guides the prediction of future events in human visual cortex and hippocampus" for consideration by *eLife*. Your article has been reviewed by 3 peer reviewers, and the evaluation has been overseen by a Reviewing Editor and Chris Baker as the Senior Editor. The following individual involved in the review of your submission has agreed to reveal their identity: Helen Barron (Reviewer #3).

Essential revisions:

The reviewers detail their essential revisions below. Our discussion converged on three key points:

1) We ask that the authors keep all model comparisons consistent across regions and tasks.

2) Additional analyses appear necessary to clarify the relationship between the hippocampus and V1.

3) In the revision it will be important to consider the successor representation model proposed here to other predictive sequence models.

*Reviewer #1 (Recommendations for the authors):*

1) If SR is the best name for the discussed model, it should be clarified why this is the case and, importantly, any difference with the SR as defined in the RL literature should be discussed. Otherwise, another term might be more appropriate.

2) It would be interesting to discuss in the discussion the distinction between the SR model and more complex models that might fit human behaviors and representations just as good or better. For example, with the current design, the SR model can't be disentangled from a more complex model in which all one-step transitions are stored and perhaps in which predictions are iteratively updated based on additional evidence (appearing items). A design in which each state is associated with multiple possible states (with different probabilities) might allow disentangling such additional possibilities.

3) There should be an additional analysis to investigate the relationship between the hippocampus and V1. I understand the limitations of fMRI and of the current experimental design, but there are still possible analyses, even if they are indirect and the results non-definitive (for example, a correlation of the hippocampal and V1 effects across individuals, as in Hindy et al., 2016, Nat Neurosci).

4) The goal of the tuning analysis and the interpretation of its result should be clarified.

5) It should be clarified whether the screen during the ITI is the same as during the omitted items of the partial sequence trials. If this is the case, the potential implications should be discussed.

6) It is unclear from the methods how the tuning analysis was performed exactly. It is a bit circular to define voxels sensitive to a given dot location based on the localizer data and then evaluate on that same data which dot representations were activated on a given trial. Was there some form of cross-validation performed? I could not find it in the code. Even if this was done correctly without double dipping, it seems strange conceptually to use the localizer data for both the fitting and testing purposes here because implicitly, the authors would both assume that the localizer data is independent of the learned associations (to determine the voxels sensitive to a given dot) and dependent on it (to assess temporal tuning). Relatedly, this somewhat applies to the other analyses too: since the localizer was performed after the main task, could it be that the authors did not select the right set, or the complete set, of voxels that are normally sensitive to a given dot location?

7) There seems to be a trend toward the last dot leading to a greater BOLD activity (Figure 3a). I'm wondering if this is because of the task, which is specific to the last dot. I don't think this explains the successor vs predecessor effect though, as you show in Figure 3c. However, this could explain the result of the current only statistical test performed in the "Anticipated stimulus sequences in V1" section. To formally exclude this possibility, the authors should test the difference in the activation of a given dot (B or C) when it is a successor vs when it is a predecessor.

8) The second important prediction of the SR model, in addition to the greater activation for successors than for predecessors, is the decreasing trend in activation for further successors. Although it is visible in the figures, it would be nice if this trend was also statistically tested and reported in the "Anticipated stimulus sequences in V1" section.

9) I don't find the time-resolved hippocampus analysis very convincing: couldn't this transient temporal profile be in response to the start of the trial rather than the missing dot (but see recommendation 5)? It would be best to perform the same analyses suggested above (recommendations 7 and 8) to really test whether the hippocampus exhibits the properties of the SR.

10) Continuing from above, concerning the time-resolved decoding: since trials are very short and ITI are jittered, it seems to me that the activity from previous trials could affect the results. Performing the decoding analysis on regression coefficients from a single-trial GLM analysis would help avoid this confound.

11) Could you show a similar figure as Figure 3c but in Figure 5 for the hippocampus? It would be helpful to see the activation related to each dot location (including the shown dot).

12) Background about predictions and predictive effects in V1 should be added to the introduction, this is currently lacking.

13) There is no mention of corrections for multiple comparisons in the paper. For example, are the tests for the significance of each item in Figure 3b corrected? This should be indicated at all relevant places in the manuscript and figure legends, along with whether the tests are one-tailed or two-tailed.

14) Concerning the model fitting analysis, I'm unsure whether the H0 model can be compared to the other two models using RMSE, since it seems to have fewer parameters. A criterion like BIC or AIC should be used in this case.

*Reviewer #2 (Recommendations for the authors):*

I had two thoughts, but I leave it to the authors to decide how to address these.

1. While I agree with the authors that this is the first evidence for SR in visual sequences (to the best of my knowledge), there is another set of studies that comes to mind looking at hippocampal contributions to sequence and duration coding of perceptual sequences, which the authors may wish to discuss:

Thavabalasingam, S., O'Neil, E. B., Tay, J., Nestor, A., and Lee, A. C. (2019). Evidence for the incorporation of temporal duration information in human hippocampal long-term memory sequence representations. Proceedings of the National Academy of Sciences, 116(13), 6407-6414.

Thavabalasingam, S., O'Neil, E. B., and Lee, A. C. (2018). Multivoxel pattern similarity suggests the integration of temporal duration in hippocampal event sequence representations. NeuroImage, 178, 136-146.

2. In the model fitting procedure, what exactly does it mean that the discount parameter γ was a free parameter (p. 18)? It would be helpful to provide a bit more clarity on this, but it's also potentially theoretically interesting in light of evidence that different neural structures represent information in line with different values of γ.

*Reviewer #3 (Recommendations for the authors):*

1. SR versus other predictive sequence models: It remains unclear to me whether the predictive activity observed in V1 is best explained by an SR model or by other models that capture predictive sequences (of which there are many). To assess whether the data is best explained by an SR model, it seems necessary to check whether two adjacent states that predict divergent future states have dissimilar representations, while two states that predict similar future states have similar representations. The data presented here is unfortunately not designed to test this comparison. Can the authors nevertheless distinguish between an SR model (e.g. Figure 4A) and a 'flat prediction' model where each stimulus predicts all possible successor states equally without any temporal discounting (i.e. A predicts B, C, and D with equal probability; B predicts C and D with equal probability but does not predict A; etc..)? It seems important to report this comparison and discuss how it may be difficult to distinguish between an SR model and a 'flat prediction' using the BOLD signal.

2. Related to point 1, it remains unclear to me why the authors consider this data to reflect an SR model, while in their previous data they characterise predictive sequences as reflecting preplay. Can the authors provide a clearer explanation for why this data is best described as an SR model rather than preplay, while Ekman et al., 2017 reflect preplay? Or do the authors consider these codes to be equivalent?

3. It is not clear to me how the ROIs are being used in Figure 3 and 4? If V1 activity reflects an SR, within a given ROI it should be possible to see evidence for backward skew in the representation of each location (consistent with Mehta et al., 2000), while at the population level there is a forward skew?

4. The authors seem to apply different models to data from different brain regions and to data from the task and localiser data. Why? For consistency and clarity would it be possible for the authors to apply the same set of models throughout, to both V1 and hippocampus, and to both task and localiser data? i.e. SR model, 'flat prediction' model, CO model, H0 model, spatial model, temporal model.

5. Related to point 4, in Figure 6 it seems that V1 data from the localiser scan does not support an SR model? This suggests that the task itself is driving the predictive sequence activity in Figures 3-4? This important difference in evidence for an SR-like code during the task and localiser scan should be emphasised and discussed.

6. How specific are these findings to V1 and hippocampus? If the authors use a searchlight analysis to look for multivariate patterns consistent with an SR model, do they not find that many brain regions show evidence for an SR representation?

7. In general, several of the reported analyses are not clearly explained. For example, how do the authors generate the reconstruction maps in Figure 2? Why was pRF mapping only performed in 7 subjects? Why were the data from the pRF maps not used to generate ROIs?

8. Statistics:

a) Can the authors clarify how they corrected for multiple comparisons when performing model comparisons?

b) The authors say they performed a one-sided t-test using data from Figure 5b. Can they clarify what they did here?

[Editors' note: further revisions were suggested prior to acceptance, as described below.]

Thank you for resubmitting your work entitled "Successor-like representation guides the prediction of future events in human visual cortex and hippocampus" for further consideration by *eLife*. Your revised article has been evaluated by Chris Baker (Senior Editor) and a Reviewing Editor.

The manuscript has been improved but there is one remaining issue that needs to be addressed, as outlined by Reviewer #1. Specifically, we thought it would be helpful to provide a bit more detail on the differences between the predictions of an SR versus model-based algorithm:

*Reviewer #1 (Recommendations for the authors):*

The authors have considerably revised their paper and they have addressed most of my comments satisfactorily. However, I remain uncertain about point 1.1.

I understand that there are no rewards in your task and that the SR algorithm can apply in the absence of rewards. I am not sure however that a model-based (MB) algorithm would make different predictions than SR in the context of your experiment. Indeed, it can be difficult to distinguish SR and MB in many contexts, especially if there is no reevaluation of the transition matrix during the experiment (Momennejad et al., 2017, Nat Hum Behav). Could the authors perhaps test what the predictions of a MB algorithm would be in their experiment (see, e.g., the equation reported in the Methods of the Momennejad paper), or otherwise explain why this would be irrelevant?

*Reviewer #2 (Recommendations for the authors):*

The authors have done a thorough job of addressing my comments. I don't have any further suggestions.

---

## [Author Response]

Essential revisions:Reviewer #1 (Recommendations for the authors):(1.1) If SR is the best name for the discussed model, it should be clarified why this is the case and, importantly, any difference with the SR as defined in the RL literature should be discussed. Otherwise, another term might be more appropriate.

We thank the reviewer for giving us the opportunity to clarify this aspect. The reviewer states that “*the SR has previously been used only in a RL context, where there are rewards associated with specific states and where predictions are task-relevant.”.* We believe that this might reflect a misunderstanding of the SR model and have revised our manuscript to point out more clearly that SR, in contrast to model-free RL algorithms, is in fact not reward dependent.

The SR learning algorithm is based on temporal-difference-learning, but instead of learning future rewards it learns discounted expected future state occupancies. This enables learning in an environment without reward and results in a representation that encodes each state in relation to its successor states, i.e. those states that are expected to be visited in the future. This aspect is now clarified in the Introduction and Discussion of our manuscript.

Introduction:

*“*In the context of hippocampal representations, the successor representation (SR) has been recently proposed (Dayan, 1993; Stachenfeld et al., 2017) to combine the trade-off between both flexible and efficient model properties. The SR postulates a predictive representation in which the current state is represented in terms of its future (successor) states, in a temporally discounted fashion. The SR is dependent on the actual experience, with states experienced more frequently being represented more strongly. This enables learning in an environment without explicit reward (Gläscher et al., 2010).2

Our discussion now also includes the following paragraph highlighting the passive nature of our task and the absence of any reward:

Discussion:

“One aspect that sets our study apart is that the viewing of the visual sequence does not require any predictive planning of the participant to evaluate different future outcomes. In contrast, related studies reporting neuronal evidence for SR-like representations in hippocampus and PFC (Barron et al., 2020; Brunec and Momennejad, 2022) and occipital cortex (Schwartenbeck et al., 2021) have used paradigms in which participants were actively engaged in prospective planning and choice evaluation. Given the relatively passive nature of our task, one might therefore wonder whether it is expected to find any map-like activity at all. However, in this context it is important to stress that the SR, unlike other model-based algorithms, does not depend on choice-dependent reward to build its transitional task structure (Momennejad et al., 2017; Stachenfeld et al., 2017) and therefore might not depend on participants’ active engagement. Furthermore, Russek et al. (2021) have recently used a paradigm in which subjects were passively exposed to transitions between visual states and reported evidence for SR-like representations in the absence of active choices in line with the results of the present study. Further supporting this notion, we have previously shown that anticipatory sequence activity occurred even after subjects’ attention was diverted from the sequence to a demanding task at fixation (Ekman et al., 2017), rendering the sequence task irrelevant. Taken together, these observations indicate that SR-like representations are not limited to situations that require active planning, or multiple-choice evaluations but may rather be formed automatically and incidentally, as has been shown repeatedly in the domain of statistical learning (Fiser and Aslin, 2002; Turk-Browne et al., 2005).”

(1.2) It would be interesting to discuss in the discussion the distinction between the SR model and more complex models that might fit human behaviors and representations just as good or better. For example, with the current design, the SR model can't be disentangled from a more complex model in which all one-step transitions are stored and perhaps in which predictions are iteratively updated based on additional evidence (appearing items). A design in which each state is associated with multiple possible states (with different probabilities) might allow disentangling such additional possibilities.

We agree with the reviewer that a different experimental design would be required to dissociate SR from other, more complex models. While there are a series of model-based algorithms that indeed store the entire transitional structure (and reward information) that can be iteratively updated, we are not aware of any such model that would align with the predictions of the SR (temporal discounting and directionality).

We have revised the manuscript Discussion to include this discussion point:

*“*For future designs it would be interesting to include visual sequences where individual states have multiple possible successor states with different probabilities associated with them. Such a design would further allow to dissociate the SR representation from alternative models that simply store all one-step transitions and their respective probabilities.”

(1.3) The second important prediction of the SR model, in addition to the greater activation for successors than for predecessors, is the decreasing trend in activation for further successors. Although it is visible in the figures, it would be nice if this trend was also statistically tested and reported in the "Anticipated stimulus sequences in V1" section.

We added a statistical test to quantify the activity decay visible in Figure 3 and revised the manuscript as follows:

*“*The activity decay toward distant future locations was formally tested by fitting an exponentially decaying factor γ γ ∈ [0,1] to each participant’s data. Here, values closer to 0 indicate a steeper decay and values closer to 1 indicate no decay. In line with our predictions, we found a group averaged decaying factor of γ = 0.14 (+/- 0.03 s.e.m.) that was statistically significantly different from 1 (non-parametric t-test t(34) = -17.17, p=2.54 × 10^-18^).”

(1.4) There seems to be a trend toward the last dot leading to a greater BOLD activity (Figure 3a). I'm wondering if this is because of the task, which is specific to the last dot. I don't think this explains the successor vs predecessor effect though, as you show in Figure 3c. However, this could explain the result of the current only statistical test performed in the "Anticipated stimulus sequences in V1" section. To formally exclude this possibility, the authors should test the difference in the activation of a given dot (B or C) when it is a successor vs when it is a predecessor.

Following the reviewer’s suggestion, we compared V1 BOLD activity across predecessor vs successor states for individual dot locations B and C. The statistical results of this control analysis replicate our main results, showing larger activity at location B during trials when B is a successor state (i.e., when dot A is presented) compared to when it is a predecessor state (i.e., when dot C is presented): t(34) = 5.72, p = 2.02 × 10^-6^. The same pattern of results was observed for location C (t(34) = 3.13, p = 0.0035), providing an internal conceptual replication. These new results show that the reported effects are also present went comparing individual dots and therefore exclude the possibility that the statistical comparison was driven by an increase in BOLD across locations.

We revised the manuscript to include the additional analysis:

*“*In line with our predictions, V1 BOLD activity was indeed enhanced at the non-stimulated successor locations compared to the non-stimulated predecessor locations (averaged across all partial trials and sequence locations; t(34) = 6.45, p = 2.23 × 10^-7^). The same pattern of future directed prediction was also evident from the visual inspection of BOLD activity for all partial sequence trials separately (Figure 3C).

Further, these results of greater activity for successor compared to predecessor activity also holds when comparing individual sequence locations without averaging (i.e., comparing non-stimulated location B when successor vs predecessor, t(34) = 5.72, p = 2.02 × 10^-6^; and location C when successor vs predecessor, t(34) = 3.13, p = 0.0035).”

(1.5) I don't find the time-resolved hippocampus analysis very convincing: couldn't this transient temporal profile be in response to the start of the trial rather than the missing dot (but see recommendation 5)? It would be best to perform the same analyses suggested above (recommendations 7 and 8) to really test whether the hippocampus exhibits the properties of the SR.

We thank the reviewer for bringing up this point. It appears that we had not properly explained what the transient profile reflects. Concretely, if the decoded information were only in response to the start of the trial, as the reviewer suggests, the time-resolved decoding profile would be completely flat.

We rephrased the result section to emphasize that the time-resolved hippocampus analysis reflects the *decoding* time-course (successor vs predecessor states) and therefore differs in its interpretation from a BOLD response. In the latter case, the transient profile could indeed reflect a bottom-up response to the starting dot, as the reviewer pointed out. However, in the case of decoding the transient profile shows that individual trials represent evidence specifically for dot locations associated with the successor states.

*“*Results of the evidence difference time-course clearly show a transient response peaking approximately 4.7 s post stimulus onset (Figure 5F) indicating that hippocampal predictions were triggered by the partial sequence dot. Note, that the decoding time-course reflects the evidence for successor locations vs predecessor locations independent of the bottom-up stimulus. The transient decoding profile can therefore not simply reflect the onset of a given trial.”

(1.6) Continuing from above, concerning the time-resolved decoding: since trials are very short and ITI are jittered, it seems to me that the activity from previous trials could affect the results. Performing the decoding analysis on regression coefficients from a single-trial GLM analysis would help avoid this confound.

We thank the reviewer for this suggestion. We would like to point out that the order of our trial sequences was counterbalanced to prevent *systematic* influences from previous trials to the current trial. We therefore believe that the single-trial GLM analysis is not strictly required in this case.

We added the motivation for counterbalancing the trial order in the revised manuscript:

*“*The pseudo-randomization (perfect counterbalancing was numerically not possible with the set number of trial types and repetitions), rules out the possibility of systematic order effects.”

2) There should be an additional analysis to investigate the relationship between the hippocampus and V1. I understand the limitations of fMRI and of the current experimental design, but there are still possible analyses, even if they are indirect and the results non-definitive (for example, a correlation of the hippocampal and V1 effects across individuals, as in Hindy et al., 2016, Nat Neurosci).

We appreciate this suggestion. To clarify, we had previously refrained from a V1-Hippocampus correlation analysis, and discussed why we believe that the results would not be very meaningful with the current design:

*“*Our study was not designed to address the question to what extent V1 and hippocampus representations are independent of each other. Here, we purposefully refrained from reporting correlations between the two regions as we could not exclude that an apparent coordination might be driven by other factors like attentional fluctuations. Future experiments, using more than one stimulus sequence could potentially address this question by comparing evidence of sequence specific representations in both areas. (p. 13)”

However, to empirically address the reviewer’s question, we correlated the averaged BOLD activity across all successor locations in V1 with the average classifier evidence across successor locations in hippocampus, across participants. No significant relationship was observed (spearman correlation, r = -0.08, p = 0.668). While there could be several reasons for this lack of relationship, we believe that a lack of power precludes us from drawing strong conclusions from this null finding. Nevertheless, for completeness, we now revised the manuscript to include this new analysis:

*“*In order to probe the relationship between hippocampus and V1 successor reactivations, we performed an across subject analysis, correlating V1 BOLD activity, averaged across all successor locations, with hippocampus classifier evidence, averaged across all successor locations. No significant relationship was observed (spearman correlation, r = -0.08, p = 0.668).”

We rephrased the discussion as follows:

*“*Our study showed no functional relationship between sequence prediction in V1 and hippocampus. However, our experimental paradigm was not primarily designed to address this question, as it does not exclude the possibility that an apparent coordination might be driven by other factors like attentional fluctuations across participants. Further, V1-hippocampus coordination might exist on a trial-by-trial level, which does not necessarily transfer to statistical comparisons across participants. Future experiments, using more than one stimulus sequence could potentially address this question by comparing evidence of sequence specific representations in both areas within participants.”

(3.1) The goal of the tuning analysis and the interpretation of its result should be clarified.

We thank the reviewer for giving us the opportunity to clarify the localizer analysis and interpretation.

We realized that the term ‘tuning’ might have been misunderstood to imply that we were quantifying the neural coding properties of a cortical region. However, this is not the case, instead we had intended to use the term ‘tuning’ to refer to a learned association, as in “after exposure hippocampus representations become tuned to a certain stimulus”.

We have now changed most occurrences of the terms ‘spatial and temporal tuning’ and replaced it with ‘spatial and temporal coactivation pattern’ to avoid this confusion. Whenever we use the term ‘tuning’, we made clear that we are talking about it in the context of learned coactivation pattern. Further, we have put bigger emphasis on the fact that we are investigating and interpreting the localizer coactivation pattern as learned associations that might persists from the main task.

Here we highlight some of these changes from the Results section:

*“*Given that we successfully trained a classifier based on eight spatial locations it might seem obvious to conclude that the underlying code for these representations is purely spatial (retinotopic) as well. This is however not necessarily the case, given that the localizer was shown after the main task and might therefore reflect persistent predictive representations. Instead, robust discrimination of sequence locations could theoretically also be based on coding of temporal properties of the sequence. Indeed, Deuker et al. (2016) have recently shown that hippocampus representations can reflect in principle both spatial and temporal aspects. In our case, a temporal coding mechanism could represent stimulus locations not based on proximity in space, but rather by proximity in time.

In order to address this question, we conducted a detailed analysis of the coactivation pattern in the stimulus localizer (Figure 6A). Note that the localizer was shown at the end of the study, allowing us to test whether learned associations persisted even after the full sequence was not relevant anymore. Here, coactivations were defined as activation of non-stimulated locations. For instance, when presenting stimulus A, locations B-C-D might become activated as well. In general, such coactivations are often attributed to noise or ambivalent responses driven by overlapping receptive fields. However, in this case we made use of the coactivation pattern to draw inferences about the learned persistent representations.”

Regarding the absence of blank screens in the localizer. The localizer did in fact have so called null-events (blank screens) where only the fixation cross was shown. Further, the presentation order of dot locations in the localizer was counterbalanced, avoiding any systematic influence of previous trials on the current trial. To answer the reviewer’s question, we don’t see how the blank screens could contribute to the pattern observed in hippocampus. Arguably, if there were any issues with the blank screen causing a certain pattern of activity, that should be visible in both V1 and hippocampus. However, the temporal coactivation pattern we describe was only observed in hippocampus, but not in V1, rendering this possibility unlikely.

(3.2) It is unclear from the methods how the tuning analysis was performed exactly. It is a bit circular to define voxels sensitive to a given dot location based on the localizer data and then evaluate on that same data which dot representations were activated on a given trial. Was there some form of cross-validation performed? I could not find it in the code. Even if this was done correctly without double dipping, it seems strange conceptually to use the localizer data for both the fitting and testing purposes here because implicitly, the authors would both assume that the localizer data is independent of the learned associations (to determine the voxels sensitive to a given dot) and dependent on it (to assess temporal tuning).

The reviewer is correct that we used leave-one-out cross-validation for the localizer analysis to prevent double dipping.

This is described in the method section:

*“*Before applying the trained classifier to the main task, we confirmed that the classifier was indeed able to distinguish between stimulus locations within the localizer. To this end, we performed a leave-one-out cross validation and tested the decoding accuracy against chance level (1/8 = 12.5 %) across subjects using a one-sample t-test. In addition to a binary classifier output for each class, we also looked at the probabilistic output. For each sample in the localizer test set, we obtained 8 probability values, one for each class. We refer to the classifier probability as classifier evidence, as the probability reflects the evidence that a particular class is represented. For each participant probability values were averaged across trials to obtain location specific response profiles.”

We apologise that the analysis code was not sufficiently documented. We previously provided analysis code that recreates the article Figures from pre-processed, intermediate data specific to each figure, and code that creates the intermediate data from the raw data. The cross-validation analysis was not included in the scripts associated with the figures. We have now improved the documentation of our analysis scripts and separated the code for the article figures from the code that processes the raw data, which makes the distinction more obvious.

Considering the aspect of ‘conceptual strangeness’, in the localizer, we are simply assessing the structure of persistent activity pattern after the main task. Our results show that no such structure is present in V1, which basically rules out any concerns. The V1 results are also confirmed using independent pRF data (see detailed response below).

The hippocampus does show a co-activation pattern, whereby not only the presented stimulus, but also other stimuli were represented, albeit to a lesser degree. Importantly, in contrast to V1, the hippocampus analysis is based on a classification analysis, that *does not* rely on a two-step process where relevant voxels are first identified, and then characterized based on their BOLD activity. Instead, the classifier takes all hippocampus voxels and outputs a probability for each possible stimulus location based on the multivariate structure.

Since the presented localizer stimulus is also the one correctly identified as most likely by the classifier (despite evidence for other stimuli), we can use the classifier to identify the presented and reactivated stimuli in the main task.

Relatedly, this somewhat applies to the other analyses too: since the localizer was performed after the main task, could it be that the authors did not select the right set, or the complete set, of voxels that are normally sensitive to a given dot location?

We thank the reviewer for bringing up the issue of correct voxel selection.

In V1, the voxel for each location were selected by (1) contrasting the BOLD activity at one location with all other locations and then (2) selecting the 25 most active voxels (highest z-value) for that location. For the main results shown in Figure 3, this contrast approach ensures that we select only voxel specific to one dot location. Even in the case of co-activation, or activity spread to neighboring locations, the contrast approach will ensure that only voxel specific to the stimulated location were selected (assuming that the region receiving the bottom-up stimulus input will always elicit the strongest BOLD response).

The location selectivity can also be empirically seen in Figure 2a (right), where we plot the BOLD activity of the selected voxel, projected into stimulus space, using independently acquired receptive-field data. Figure 2a (right) shows that the selected voxels were indeed at the expected stimulus location, and not at other receptive field locations in the visual field.

We have now stressed more clearly that the pRF data in Figure 2 validate that we selected the right set of voxels for a given dot location:

*“*Stimulus response profiles of these eight (retinotopic) ROIs show little coactivation of neighboring locations in the visual field which allows for a precise investigation of location specific activity (Figure 2A). Unsurprisingly, during full sequence trials BOLD activity at the sequence locations receiving bottom-up visual input was markedly enhanced compared to non-stimulated control locations (Figure 2B). Population-based receptive field (pRF) data, that was acquired for a subset of participants confirmed that the selected voxels correspond to the retinotopic stimulus locations as expected.”

For the additional results shown in Figure 6, we show data from a *baseline contrast* (opposed to the direct contrast employed for the main results).

4) It should be clarified whether the screen during the ITI is the same as during the omitted items of the partial sequence trials. If this is the case, the potential implications should be discussed.

The reviewer is correct that there is no visual difference between the inter-trial interval (ITI) and the part of the sequence where no dot is shown. By design, the variable ITI prevents the subject from learning any temporal structures related to the start of a trial. In doing so, we can focus on the predictive process that is triggered by the presentation of a sequence dot, independent of any temporal expectation effects.

In the revised manuscript, we have now clarified that the variable ITI looks visually identical to the omission of sequence dots and therefore ensures that we are not confounding the predictive effects of interest with any temporal expectation effects.

“The variable ITI ensured that the experimental paradigm had no temporal structure that participants could learn to expect the onset of a trial. This allowed us to focus in the present study on the learning and representation of structural knowledge, independent of any temporal expectation effects.

To probe activity replay we introduced partial sequence trials where only one of the four dots was shown for 100 ms, instead of the full sequence. Visually, there was no difference between the ITI and the part of the partial sequence trials where the dots were omitted, both showed a fixation cross at the center of the screen.”

5) Could you show a similar figure as Figure 3c but in Figure 5 for the hippocampus? It would be helpful to see the activation related to each dot location (including the shown dot).

Given the significant, but very low classification accuracy in within the localizer (accuracy = 15% 3.6%, mean ± s.d.; p = 0.002), we had previously decided to only report averaged location results for the hippocampus as the non-averaged predictions would be very noisy. To put the hippocampus classification accuracy into context, in V1 cross-validated accuracy within the localizer was (92% ± 12%, mean ± s.d.).

We now stressed this difference between V1 and hippocampus decoding in the Results section and motivate our reason for presenting averaged results:

*“*Within localizer decoding accuracy results confirmed that hippocampus has a coarse representation of the eight stimulus locations (Figure 5B) within the localizer (one-sample t-test; t(34) = 3.28, p = 0.002; cross-validated accuracy = 15% ± 3.6%, mean ± s.d.; see *Materials and methods*). Notably, compared to V1 (cf. Figure 2A), within localizer accuracy was relatively low and as a consequence tuning curves in hippocampus appeared less sharp (Figure 5C). In order to maximize sensitivity for the hippocampus, we averaged classification evidence across successor and predecessor locations. Non-averaged results can be found in Supplementary Figure 1A.”

Further, we followed the reviewer’s suggestion and added a new supplementary Figure including the non-averaged results for hippocampus. The new Figure also includes the model comparison the reviewers had asked for.

6) Background about predictions and predictive effects in V1 should be added to the introduction, this is currently lacking.

We rephrased the introduction to focus more on predictive effects in V1:

*“*Previous research has repeatedly shown that prior expectations influence neural activity in the visual cortex (Ekman et al., 2017; Gavornik and Bear, 2014; Hindy et al., 2016; Kok et al., 2012; Xu et al., 2012). It remains, however, unknown if SR-like representations are present outside the hippocampus in areas like the early visual cortex (V1) that have a strong retinotopic organization. Theoretically it is possible that V1 receptive fields, analogous to hippocampal place fields, become tuned to respond not only to the current input, but also to expected future inputs. Here we propose that the computationally efficient and flexible properties of the SR could in theory also underlie the anticipation of future events in V1.”

7) There is no mention of corrections for multiple comparisons in the paper. For example, are the tests for the significance of each item in Figure 3b corrected? This should be indicated at all relevant places in the manuscript and figure legends, along with whether the tests are one-tailed or two-tailed.

We thank the reviewer for pointing this out. We added the information to the legend of Figure 3, Figure 4 and Figure 6:

*“*Error bars denote ± s.e.m.; two-tailed t-test, ***P<0.001; **P<0.01; *P<0.05 uncorrected for multiple comparisons.”

8) Concerning the model fitting analysis, I'm unsure whether the H0 model can be compared to the other two models using RMSE, since it seems to have fewer parameters. A criterion like BIC or AIC should be used in this case.

We implemented this suggestion and calculated BIC, instead of RMSE for every subject, thereby controlling for the difference in model parameters. The results remain unchanged compared to the previous version of the manuscript. In short, the SR model has the smallest BIC value (smaller = more likely), followed by the CO model and the H0 model.

We updated Figure 4c, Figure 6f and the related Results and Method section.

Reviewer #2 (Recommendations for the authors):I had two thoughts, but I leave it to the authors to decide how to address these.1. While I agree with the authors that this is the first evidence for SR in visual sequences (to the best of my knowledge), there is another set of studies that comes to mind looking at hippocampal contributions to sequence and duration coding of perceptual sequences, which the authors may wish to discuss:Thavabalasingam, S., O'Neil, E. B., Tay, J., Nestor, A., and Lee, A. C. (2019). Evidence for the incorporation of temporal duration information in human hippocampal long-term memory sequence representations. Proceedings of the National Academy of Sciences, 116(13), 6407-6414.Thavabalasingam, S., O'Neil, E. B., and Lee, A. C. (2018). Multivoxel pattern similarity suggests the integration of temporal duration in hippocampal event sequence representations. NeuroImage, 178, 136-146.

We thank the reviewer for pointing out these articles. We have now included both references in the revised manuscript.

*“*Hippocampus on the other hand represented relevant items predominantly in terms of their temporal distance within the sequence, suggesting that representations capitulate on the transitionally structure of the visual sequence. These results align with previous reports that hippocampus can learn to represent temporal sequence structure (Thavabalasingam et al., 2018, 2019) and temporal proximity in a spatial navigation task (Deuker et al., 2016; Howard et al., 2014), but to the best of our knowledge, constitute the first reports of coding temporal distance of a visual sequence.”

2. In the model fitting procedure, what exactly does it mean that the discount parameter γ was a free parameter (p. 18)? It would be helpful to provide a bit more clarity on this, but it's also potentially theoretically interesting in light of evidence that different neural structures represent information in line with different values of γ.

Keeping γ as a „free parameter” was meant to convey, that instead of using a fixed value for γ (e.g., based on previous literature) and fitting the curve to all participants, the value of γ was determined during data fitting for each participant individually. We rephrased this formulation to make that clearer.

*“*During model fitting γ was a free parameter, meaning that instead of using a fixed value, individual γ values were determined for each participant. Here, larger values of γ result in a smaller exponential decay of future states.”

We also report group statistics of obtained γ values:

*“*The activity decay toward distant future locations was formally tested by fitting an exponentially decaying factor γ γ ∈ [0,1] to each participant’s data. Here, values closer to 0 indicate a steeper decay and values closer to 1 indicate no decay. In line with our predictions, we found a group averaged decaying factor of γ = 0.14 (+/- 0.03 s.e.m.) that was statistically significantly different from 1 (non-parametric t-test t(34) = -17.17, p=2.54 × 10^-18^).”

Reviewer #3 (Recommendations for the authors):1. SR versus other predictive sequence models: It remains unclear to me whether the predictive activity observed in V1 is best explained by an SR model or by other models that capture predictive sequences (of which there are many). To assess whether the data is best explained by an SR model, it seems necessary to check whether two adjacent states that predict divergent future states have dissimilar representations, while two states that predict similar future states have similar representations. The data presented here is unfortunately not designed to test this comparison. Can the authors nevertheless distinguish between an SR model (e.g. Figure 4A) and a 'flat prediction' model where each stimulus predicts all possible successor states equally without any temporal discounting (i.e. A predicts B, C, and D with equal probability; B predicts C and D with equal probability but does not predict A; etc.)? It seems important to report this comparison and discuss how it may be difficult to distinguish between an SR model and a 'flat prediction' using the BOLD signal.

The reviewer points out that there are many other possible predictive activity patterns that could be expected. We are actually not aware of any existing (biological) model that would generate predictions, selectively for successor states and not for predecessor states.

While we agree with the reviewer’s point that there are many possible predictive activity *patterns*, like ‘flat prediction’, ‘linear decrease prediction’, ‘linear increase predictions’, to the best of our knowledge, none of these predictions can be derived from existing *models*. That’s why we had previously only included one alternative model, the co-occurrence model which is based on a biological framework in which autoassociative connections within the hippocampal CA3 regions reactivate related sequence items from partial input, without skewing toward future locations. To the best of our knowledge, the SR model is the only model that predicts an asymmetry toward future locations.

While we like to keep the focus of our manuscript on the two existing, biologically motivated models, we have calculated the suggested ‘flat prediction’ model for the revision letter.

*“*Comparing the model fit (BIC) of the suggested ‘flat prediction’ pattern with the SR model showed that the SR model describes the data significantly better (two-sided t-test, t(34) = 6.12, p = 5.98 x 10^-7^).”

2. Related to point 1, it remains unclear to me why the authors consider this data to reflect an SR model, while in their previous data they characterise predictive sequences as reflecting preplay. Can the authors provide a clearer explanation for why this data is best described as an SR model rather than preplay, while Ekman et al., 2017 reflect preplay? Or do the authors consider these codes to be equivalent?

Previous to the present study we didn’t know whether the observed preplay/replay traces were guided by a generative model that represents the relational structure of the environment. Our previous paradigm in Ekman et al. 2017 was not designed to address this question, as the dot sequence (i) had no intermediate omissions and (ii) the dot locations have different eccentricities from fixation which hinders the interpretation of the absolute BOLD values.

The difference with our previous study is discussed as follows:

*“*There is an extensive body of literature that shows how expectations elicit anticipatory activity in early visual cortices (de Lange et al., 2018; Hindy et al., 2016; Kok et al., 2012). For instance, we have previously shown that flashing an individual dot of a simple, linear sequence triggers an activity wave in V1 that resembles the full stimulus sequence (Ekman et al., 2017, 2022), akin to replay of place field activity during spatial navigation (Foster and Wilson, 2006; Gupta et al., 2010). However, what remains unknown is whether these sensory replay traces are guided by a generative model that represents the relational structure of the stimulus sequence, akin to a predictive map. Alternatively, anticipatory activity traces could simply reflect the association between different stimuli, based on their co-occurrence, without the added complexity of any temporal relational structure. The latter explanation appears plausible, given that predictive representations in early visual cortex are generally time critical and operate in parallel to a constant stream of new sensory input, which arguably requires efficient processing and in turn limits the complexity of such representations.

In fact, we previously speculated that cue-triggered reactivation of simple sequences might be driven by an automatic pattern completion-like mechanism that reactivates all associated items based on partial input (Ekman et al., 2017). This idea is in line with the finding that predictive representations in V1 correlated with pattern completion-like activity in the hippocampus (Hindy et al., 2016; Kok and Turk-Browne, 2018) that might be driving V1 activity (Finnie et al., 2021; Ji and Wilson, 2007).

Our current findings directly challenge this interpretation and instead point to a predictive representation of expected, temporally discounted, future states. We accomplished this by using a paradigm in which one visual event (e.g., the presentation of one dot) was framed as one state in a directed transition matrix with a fixed relational structure. The SR hypothesis makes two testable predictions, namely that population activity represents future states over predecessor states, and that future state representations are temporally discounted, such that events in the close future are more prominently represented compared to events in the distant future. Using a paradigm in which we occasionally presented only single items of the full sequence, allowed us to investigate V1 activity at omitted sequence locations.”

3. It is not clear to me how the ROIs are being used in Figure 3 and 4? If V1 activity reflects an SR, within a given ROI it should be possible to see evidence for backward skew in the representation of each location (consistent with Mehta et al., 2000), while at the population level there is a forward skew?

We believe that this is indeed what our data shows. For example, within the V1 ROI that is responsive to dot location B, there is elevated activity when dot A is shown. This could be interpreted as ‘backward skew’ of this ROI: the ROI that is tuned to location B also starts responding to location A.

At the level of the entire V1 population, however, this results in “forward skew”: when presenting dot location A, the population response is skewed forward, by virtue of the anticipatory activity in V1 neurons that are tuned to successor location B.

4. The authors seem to apply different models to data from different brain regions and to data from the task and localiser data. Why? For consistency and clarity would it be possible for the authors to apply the same set of models throughout, to both V1 and hippocampus, and to both task and localiser data? i.e. SR model, 'flat prediction' model, CO model, H0 model, spatial model, temporal model.

We appreciate the suggestion made by the reviewer to apply the same models to both V1 and hippocampus.

For the hippocampus, we had previously analysed averaged classifier outputs across locations. This was done to effectively improve the signal-to-noise ratio. However, averaging the output (i.e., all successor locations vs all predecessor locations), did not allow us to do any model fitting. In the revised version, we have now implemented six changes: (1) we added our motivation for collapsing the hippocampus data (2) we now show the non-averaged hippocampus results as a Supplementary Figure (3) we report the same model comparison for hippocampus that was done for V1, thereby keeping the model comparison consistent across regions (4) we now include the SR model in the model comparison for the localizer (5) we added our motivation for applying the spatial and temporal models to the localizer and not to the main task. (6) we renamed the no coactivation (NoCo) model from the localizer to H0 model, indicating more clearly that this is the same ‘baseline’ model used in the main task. The different names (H0, NoCo) might have previously contributed to the impression that these are different models, despite being conceptually the same.

Below we copy our response to Reviewer #1 from above, who brought up a similar point.

Given the significant, but very low classification accuracy in within the localizer (accuracy = 15% 3.6%, mean ± s.d.; p = 0.002), we had previously decided to only report averaged location results for the hippocampus as the non-averaged predictions would be very noisy. To put the hippocampus classification accuracy into context, in V1 cross-validated accuracy within the localizer was (92% ± 12%, mean ± s.d.).

We no stressed this difference between V1 and hippocampus decoding in the Results section and motivate our reason for presenting averaged results:

*“*Within localizer decoding accuracy results confirmed that hippocampus has a coarse representation of the eight stimulus locations (Figure 5B) within the localizer (one-sample t-test; t(34) = 3.28, p = 0.002; cross-validated accuracy = 15% ± 3.6%, mean ± s.d.; see *Materials and methods*). Notably, compared to V1 (cf. Figure 2A), within localizer accuracy was relatively low and as a consequence tuning curves in hippocampus appeared less sharp (Figure 5C). In order to maximize sensitivity for the hippocampus, we averaged classification evidence across successor and predecessor locations. Non-averaged results can be found in Supplementary Figure 1A.”

Further, we followed the reviewer’s suggestion and added a new supplementary Figure including the non-averaged results for hippocampus. The new Figure also includes the model comparison the reviewers had asked for.

Additionally, we also followed the reviewer’s suggestion and included the SR model to the localizer analysis, confirming that the localizer is not best described by an SR coactivation pattern.

Finally, we explained why we don’t apply the temporal and spatial coactivation models to the main task. Here we copy our reply to Reviewer #2 from above, who had a similar point:

The reviewer is correct that the fact that the sequence order and spatial distance were not fully decorrelated (second presentation was always farthest away from starting dot, third and fourth dot always the same distance from start) prevents us from quantifying the interaction of the SR and CO model with a spatial model during the main task.

We added the following to the Method section to clarify this:

*“*Note that because within each dot sequence, temporal order and spatial distance were not perfectly decorrelated (e.g. the second sequence dot was always farthest apart from the starting dot), it is not possible to estimate the combined influence of the SR model and the spatial coactivation model on the observed BOLD activity.”

Having said that, we believe that there is little concern that the reported reactivations of the main task are driven by the Euclidean distance in a meaningful way for two reasons:

(1) Detailed analysis of the localizer data showed that there is no spatial spreading from one dot location to the other sequence locations (Figure 6). This is likely because the relevant dot locations were sufficiently spaced apart. Given the lack of spreading during the localizer, where the dot was flashed for 13.5s, makes the presence of spreading during the main task, where the dot was flashed for only 100ms, equally unlikely.

(2) The presence of spatial spreading would actually obfuscate the reported SR-like pattern and could not have caused it. Specifically, because the second sequence dot was always farthest apart from the start, this is where one would assume the least amount of activity spread (greatest Euclidean distance). Sequence dots three and four should be more active given that they are both closer to the starting point in terms of Euclidean distance. Our reported results are the opposite of that pattern, ruling out the possibility that these were caused by spatial spreading.

5. Related to point 4, in Figure 6 it seems that V1 data from the localiser scan does not support an SR model? This suggests that the task itself is driving the predictive sequence activity in Figures 3-4? This important difference in evidence for an SR-like code during the task and localiser scan should be emphasised and discussed.

The reviewer is correct that V1 data from the localizer does not show the persistent SR-like predictions from the main task. In the revised version, we now included a formal test for this (see response above).

We believe that this is to be expected as the sequence predictions are learned and updated based on exposure. Within the localizer, no more dot sequences are shown. Instead, individual dot locations are repeatedly flashed for 13.5 s at the same location. It is therefore expected that sequence predictions, related to the previous task, would eventually fade away. This can be understood in the context of continuous updating of the predictions once the regularities of the environment change and does not constitute any evidence against the SR model.

We have added the following to the Discussion:

*“*Furthermore, hippocampus predictive codes were found to persist after the sequence task and coactivation of related sequence locations were still present during the stimulus localizer, potentially indicating that hippocampus representations reflect a more stable code operating on a longer timescale. V1 representations on the other hand did not persist throughout the stimulus localizer and reverted back to representing individual spatial locations without coactivation of related sequence locations, further highlighting another qualitative difference between V1 and hippocampus coding. According to the SR, it is expected that sequence predictions will change once the regularities of the environment change. The absence of SR-like pattern in V1 during the functional localizer is therefore not at odds with our results from the main task, but rather indicative of a dynamic updating of the generative model.”

6. How specific are these findings to V1 and hippocampus? If the authors use a searchlight analysis to look for multivariate patterns consistent with an SR model, do they not find that many brain regions show evidence for an SR representation?

We addressed the reviewer’s question about the specificity of the effects, by testing another low-level visual area V2. These new results show that in contrast to V1 and hippocampus, V2 does not feature any predictive effects and suggest that the reported findings are not ubiquitous throughout the brain.

This new result is mentioned in the revised manuscript:

*“*One could ask whether our findings are specific to V1 and hippocampus, or widespread throughout the brain. In order to answer this question, we repeated the analysis for low-level visual area V2. In contrast to V1, no predictive effects were found in area V2. V2 BOLD activity was not enhanced at the non-stimulated successor locations compared to the non-stimulated predecessor locations (averaged across all partial trials and sequence locations; t(34) = 1.41, p = 0.168).”

We appreciate the searchlight suggestion to complement our ROI analysis approach. However, we believe that an additional ROI analysis in this case is more meaningful compared to a searchlight analysis. The reason for this is that the dot locations in our experiment are up to ~14 degrees apart in visual space. Within retinotopically organized visual areas, these stimulus locations are represented in different hemifields, multiple centimetres away.

The sphere of a searchlight with a commonly used radius of ~4mm would not be able to capture that effect, simply because the sphere would be too small to include all relevant voxel. One could argue that running a searchlight analysis with a radius of ~40mm (a magnitude larger) would alleviate that problem, but that would result in a complete loss of spatial specificity. For this reason, we refrained from using a searchlight analysis and present the additional V2 results instead.

7. In general, several of the reported analyses are not clearly explained. For example, how do the authors generate the reconstruction maps in Figure 2? Why was pRF mapping only performed in 7 subjects? Why were the data from the pRF maps not used to generate ROIs?

We thank the reviewer for pointing out these unclarities. In short, the pRF data were available for 7 subjects, that had also participated in a previous experiment. Since the goal of the pRF mapping was only to confirm that the localizer voxel-selection resulted in accurate results (Figure 2), we decided not to invite all subjects for a second pRF session that would have taken 45 minutes of fMRI scanning time.

The ROIs were generated from the functional localizer, because the pRF data was not available for all subjects. Please note that the voxel selection via the functional localizer was also the method we had preregistered for the data analysis (https://osf.io/f8dv9/), so this was always the intended analysis.

In the revised manuscript we now stress, that the pRF data was from a previous experiment:

*“*pRF estimation*.* pRF data was available for seven participants from a previous study and was used to validate visually that the voxel selection based on the functional localizer selected voxel that correspond to the stimulated location is visual space (Figure 2).”

We further added the missing information how the pRF reconstruction in Figure 2 was performed:

“For the pRF-based stimulus reconstruction, for each participant, we first limited the pRF data to voxel that were selected based on the functional localizer (25 voxel per stimulus location, 200 voxels in total). This selection step would allow us to visually inspect whether the voxel selection accurately selected voxel corresponding to the respective stimulus location. Second, every voxel is described as a 2D Gaussian with parameters x0, y0, and s0 from the pRF estimation. The 2D Gaussians for each voxel, represented by a pixel x pixel image, were scaled based on the percent signal change obtained from the functional localizer GLM, and consecutively summed over voxels to create one 2D representation of the reconstructed stimulus. This procedure was repeated separately for all 8 stimulus locations. Finally, for visualization purpose, the 8 individual localizer conditions were rotated to one stimulus location (22.5°) and averaged across stimulus locations and participants.”

8. Statistics:a) Can the authors clarify how they corrected for multiple comparisons when performing model comparisons?

Thank you for pointing this out. No correction for multiple comparisons was carried out during the model comparison. This was now added to the legend of Figure 6.

*“*Error bars denote ± s.e.m.; two-tailed t-test, ***P<0.001; **P<0.01; *P<0.05 uncorrected for multiple comparisons.”

b) The authors say they performed a one-sided t-test using data from Figure 5b. Can they clarify what they did here?

We apologize for the unclarity. We performed a *two-sided* one-sample t-test. We revised the paragraph in the main text to refer to the Methods section where the analysis is explained:

“Within localizer decoding accuracy results confirmed that hippocampus has a coarse representation of the eight stimulus locations (Figure 5B) within the localizer (two-sided one-sample t-test; t(34) = 3.28, p = 0.002; cross-validated accuracy = 15% ± 3.6%, mean ± s.d.; see *Materials and methods*).”

The Methods section under “Hippocampal decoding” reads:

“A logistic regression classifier (default values, L2 regularization; C=1) was trained to distinguish between 8 stimulus locations during the independent localizer run. Before applying the trained classifier to the main task, we confirmed that the classifier was indeed able to distinguish between stimulus locations within the localizer. To this end, we performed a leave-one-out cross validation and tested the decoding accuracy against chance level (1/8 = 12.5 %) across subjects using a one-sample t-test.”

[Editors' note: further revisions were suggested prior to acceptance, as described below.]

The manuscript has been improved but there is one remaining issue that needs to be addressed, as outlined by Reviewer #1. Specifically, we thought it would be helpful to provide a bit more detail on the differences between the predictions of an SR versus model-based algorithm:Reviewer #1 (Recommendations for the authors):The authors have considerably revised their paper and they have addressed most of my comments satisfactorily. However, I remain uncertain about point 1.1.I understand that there are no rewards in your task and that the SR algorithm can apply in the absence of rewards. I am not sure however that a model-based (MB) algorithm would make different predictions than SR in the context of your experiment. Indeed, it can be difficult to distinguish SR and MB in many contexts, especially if there is no reevaluation of the transition matrix during the experiment (Momennejad et al., 2017, Nat Hum Behav). Could the authors perhaps test what the predictions of a MB algorithm would be in their experiment (see, e.g., the equation reported in the Methods of the Momennejad paper), or otherwise explain why this would be irrelevant?

We thank the reviewer for raising this issue, which has prompted us to reflect more thoroughly on this issue.

The reviewer is correct that, within the context of our design, it is strictly speaking not possible to distinguish between model-based (MB) and SR algorithms. The key distinction between them is that SR caches a predictive map of states that the agent expects to visit in the future, whereas MB algorithms store a full model of the world and compute trajectories at the decision time. Both predict a temporally discounted activation of successor states.

It should be noted however that MB comes at a higher computational cost, and is more intensive both in terms of time and working memory resources. The activation of successor states that we observed occurred in the absence of a decision-making process (i.e., participants did not perform any task on the trials where a single dot was presented). Also, and interestingly, we previously observed that this activation pattern was not dependent on the task, and was equally present when attentional resources were strongly drawn away from the stimuli (Ekman et al. Nat Comm 2017). These observations may be more readily in line with the automatic (cached) activation of successor states that is embodied by SR, rather than the effortful iterative calculation of successor states that is the hallmark of MB. Nevertheless, we agree with the reviewer that our study does not provide strong evidence in favor of SR over MB computations in the visual cortex. We have now made this clearer in the Discussion section of our manuscript (see below).

Also, the question raised by the reviewer inspired a potential follow-up experiment, which is outside of the scope of the current manuscript, but which would be a potentially promising avenue of future research. The transition revaluation manipulation described earlier (Momennejad et al. 2017) could also be applied to our experimental setting. After exposing participants to our dot sequences (ABCD), one could introduce a relearning phase in which participants are exposed to BDC. When participants, after this relearning phase, are exposed to A, there are competing predictions about the activation pattern of successor states: H1 (SR): BCD

H2 (MB): BDC

We believe that this could be an interesting follow-up experiment, and have added it to the Discussion section of the manuscript.

We have added the following section to the Discussion section (page 13):

*“*While we have interpreted the neural activity patterns in the light of the SR, it is strictly speaking not possible to distinguish between model-based (MB) and SR algorithms within the context of our design. The key distinction between them is that SR caches a predictive map of states that the agent expects to visit in the future, whereas MB algorithms store a full model of the world and compute trajectories at the decision time (Mommenejad et al. 2017, Gershman 2018). Therefore, both predict a temporally discounted activation of successor states. It should be noted however that MB comes at a higher computational cost, and is more intensive both in terms of time and working memory resources. The activation of successor states that we observed, on the other hand, occurred in the absence of a decision-making process (i.e., participants did not perform any task on the trials where a single dot was presented). Also, importantly, we previously observed that this activation pattern was not dependent on the task, and was equally present when attentional resources were strongly drawn away from the stimuli (Ekman et al., 2017). These observations may be more readily in line with the automatic (cached) activation of successor states that is embodied by SR, rather than the effortful iterative calculation of successor states that is the hallmark of MB. One future possibility to disentangle SR and MB algorithms could be to probe how well each model adapts to changes in the dot sequence structure. It has previously been shown, that compared to MB, the flexibility of the SR is somewhat limited to reflect changes in the transitional structure, because it requires the entire SR to be relearned (Momennejad et al., 2017).”